# RewardMap: Tackling Sparse Rewards in Fine-grained Visual Reasoning via Multi-Stage Reinforcement Learning

**Sicheng Feng**[1,†], **Kaiwen Tuo**[1,2,†], **Song Wang**[3], **Lingdong Kong**[4], **Jianke Zhu**[3], **Huan Wang**[1,*]

[1]Westlake University    [2]Tongji University    [3]Zhejiang University
[4]National University of Singapore

[†]Equal contributions.    [*]Corresponding author.

🤗 **Dataset & Toolkit:** https://fscdc.github.io/RewardMap

## Abstract

Fine-grained visual reasoning remains a core challenge for multimodal large language models (MLLMs). The recently introduced REASONMAP highlights this gap by showing that even advanced MLLMs struggle with spatial reasoning in structured and information-rich settings such as transit maps, a task of clear practical and scientific importance. However, standard reinforcement learning (RL) on such tasks is impeded by sparse rewards and unstable optimization. To address this, we first construct REASONMAP-PLUS, an extended dataset that introduces dense reward signals through Visual Question Answering (VQA) tasks, enabling effective cold-start training of fine-grained visual understanding skills. Next, we propose REWARDMAP, a multi-stage RL framework designed to improve both visual understanding and reasoning capabilities of MLLMs. REWARDMAP incorporates two key designs. **First**, we introduce a difficulty-aware reward design that incorporates detail rewards, directly tackling the sparse rewards while providing richer supervision. **Second**, we propose a multi-stage RL scheme that bootstraps training from simple perception to complex reasoning tasks, offering a more effective cold-start strategy than conventional Supervised Fine-Tuning (SFT). Experiments on REASONMAP and REASONMAP-PLUS demonstrate that each component of REWARDMAP contributes to consistent performance gains, while their combination yields the best results. Moreover, models trained with REWARDMAP achieve an average improvement of 3.47% across 6 benchmarks spanning spatial reasoning, fine-grained visual reasoning, and general tasks beyond transit maps, underscoring enhanced visual understanding and reasoning capabilities.

## 1 Introduction

Fine-grained visual reasoning over structured visual inputs remains a significant challenge for multimodal large language models (MLLMs) (Bai et al., 2025b; OpenAI, 2025; Zhang et al., 2025c; Liang et al., 2025; Li et al., 2025c;d). Recently, REASONMAP (Feng et al., 2025b) was introduced as a benchmark on high-resolution transit maps in the real world, where tasks (*e.g.*, route planning) combine visual understanding with spatial reasoning, jointly constituting the fine-grained visual reasoning challenge (Xie et al., 2025a; Hu et al., 2025b; Kong et al., 2025b;a). This task is not only of practical value for real-world navigation and transportation systems, but also of fundamental scientific interest as it exposes reasoning gaps in current MLLMs. Despite steady advances in vision–language pre-training (Liu et al., 2023; 2024; Bai et al., 2025b), existing models consistently struggle with the visual and spatial reasoning demands in REASONMAP. This gap motivates us to investigate how reinforcement learning (RL) (Zhang et al., 2025a; Shao et al., 2024; Guo et al., 2025a) can be adapted to enhance fine-grained visual reasoning abilities in structured visual domains such as transit maps.

However, directly applying standard RL methods (Rafailov et al., 2023; Shao et al., 2024) to complex tasks such as REASONMAP is highly challenging, as supervision signals are inherently sparse, *i.e.*,

success is typically judged only at the final answer after a long reasoning chain (Quadros et al., 2025; Cao et al., 2024; Chen et al., 2025). The difficulty of the tasks further amplifies this sparsity, which in turn destabilizes optimization and hinders effective exploration (Wang et al., 2025a). While classical approaches like Supervised Fine-Tuning (SFT) (Liu et al., 2023; Wei et al., 2025) offer dense supervision, they fall short in equipping models for the long-chain decision-making intrinsic to visual reasoning tasks. This mismatch between task complexity and supervision signal forms a critical bottleneck in leveraging RL for fine-grained visual reasoning.

To address this issue, we first construct REASONMAP-PLUS, an extended dataset that introduces dense reward signals for further cold-start training. Tasks in REASONMAP-PLUS are organized along a natural difficulty continuum, from simple Visual Question Answering (VQA) that enhances perception to progressively harder visual tasks reflecting the complexity of fine-grained visual reasoning queries. We further introduce REWARDMAP, a multi-stage RL framework with detailed reward design. It consists of two key components: (1) a reward scheme that, beyond basic format and correctness rewards, incorporates detail rewards to mitigate sparsity in supervision for hard samples and adopts a difficulty-aware design to account for task complexity; and (2) a cold-start strategy that departs from SFT-based initialization (Xu et al., 2024; Guo et al., 2024; Wei et al., 2025; Yan et al., 2025) by directly employing RL, ensuring alignment between reward signals and task objectives from the outset. Training data are organized from easy to hard across multiple stages, with dense and accessible rewards at lower levels supporting effective cold-start training. This staged strategy systematically bridges perception and reasoning within a unified RL framework.

We conduct extensive experiments on REASONMAP and REASONMAP-PLUS to evaluate the effectiveness of REWARDMAP. Results indicate that each component yields measurable gains, with their integration delivering the best overall performance. Moreover, beyond the targeted benchmarks, models trained with REWARDMAP achieve consistent improvements (3.47% average) across six benchmarks (Wu & Xie, 2024; Wang et al., 2024a; Li et al., 2024; Wang et al., 2025f; Masry et al., 2022; Chen et al., 2024a) covering spatial reasoning, fine-grained visual reasoning, and general tasks, suggesting enhanced general visual perception and reasoning capabilities.

In summary, this work makes the following contributions: **(1)** We introduce REASONMAP-PLUS, an extended dataset organized from easy to hard, providing dense supervision for multi-stage RL training; **(2)** We propose REWARDMAP, a multi-stage RL framework that integrates cold-start curriculum data (*i.e.*, easy → hard) with difficulty-aware detail reward design; **(3)** Extensive experiments demonstrate that REWARDMAP not only improves performance on REASONMAP and REASONMAP-PLUS but also enhances performance across broader visual benchmarks beyond transit maps. Together, these contributions establish a principled approach to overcoming sparse reward challenges in visual reasoning, advancing the capabilities of MLLMs in structured visual tasks.

## 2 RELATED WORK

**Visual Reasoning in MLLMs.** The development of MLLMs has rapidly progressed from foundational models that bridge vision and language encoders, such as Flamingo (Alayrac et al., 2022), to those enhanced by visual instruction tuning like LLaVA (Liu et al., 2023; 2024). To elicit more complex, step-by-step reasoning, researchers have adapted Chain-of-Thought (CoT) (Wei et al., 2022) prompting from the language domain to the multimodal context (MCoT) (Zhang et al., 2023). However, a key limitation of early MCoT is its reliance on purely textual rationales (Wang et al., 2025g), which can be a bottleneck for expressing fine-grained visual logic (Zhang et al., 2025c). More recent work has thus focused on developing vision-centric reasoning processes (Dong et al., 2025; Man et al., 2025), such as generating intermediate visual representations as perception tokens to aid the reasoning chain (Bigverdi et al., 2025). Despite these advances, a significant performance gap persists. Benchmarks specifically designed to test abstract, spatial, and logical reasoning, such as VisuLogic (Xu et al., 2025) and REASONMAP (Feng et al., 2025b), reveal that even state-of-the-art models struggle with tasks that require high-fidelity visual and topological understanding, underscoring the need for new methods tailored for structured visual domains, such as transit maps.

**Reinforcement Learning for Reasoning.** Reinforcement Learning (RL) (Wu et al., 2025a; Sarch et al., 2025; Feng et al., 2025a) has emerged as a powerful paradigm for improving the reasoning capabilities of multimodal and language-only models beyond the static nature of Supervised Fine-Tuning (SFT) (Liu et al., 2023; Wei et al., 2025). The evolution of RL for reasoning spans from

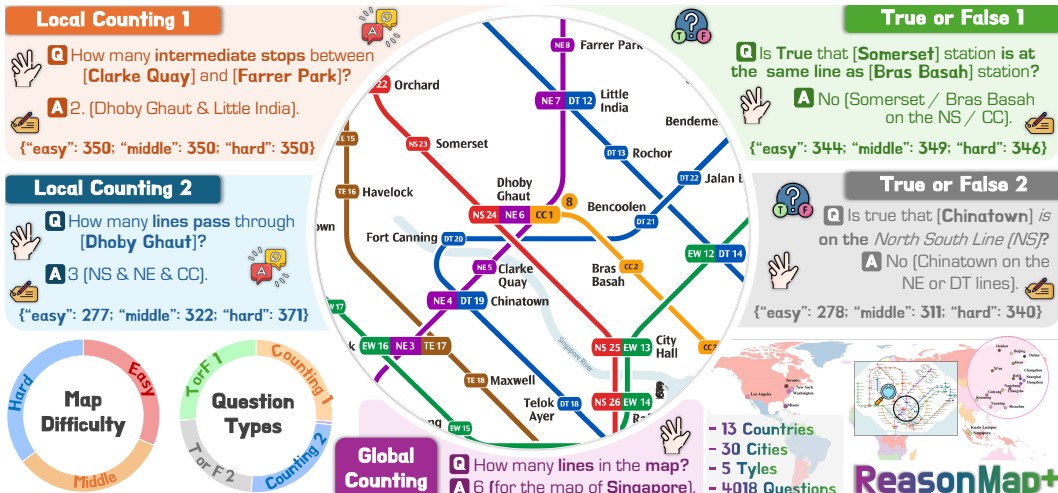

Figure 1: Overview of REASONMAP-PLUS. REASONMAP-PLUS comprises 4,018 questions from 5 extended question types and maps from 30 cities across 13 countries.

traditional RLHF pipelines (Bai et al., 2022) to more stable and direct policy optimization objectives such as Direct Preference Optimization (DPO) (Rafailov et al., 2023) and Group Relative Policy Optimization (GRPO) (Shao et al., 2024). Recent work further demonstrates the effectiveness of structured or curriculum-based RL strategies for enhancing LLM reasoning. Curriculum RL (Parashar et al., 2025) shows that progressing from easy to hard tasks can significantly strengthen reasoning robustness; Kimi K1.5 (Team et al., 2025) adopts multi-stage RL to improve long-chain and tool-integrated reasoning; and Logic-RL (Xie et al., 2025b) leverages logical structure to guide policy optimization. While SFT on CoT-rich datasets can activate latent reasoning abilities (Xu et al., 2024; Guo et al., 2024; Wei et al., 2025), it often leads to overfitting and cognitive rigidity (Chu et al., 2025; Shen et al., 2025; Wang et al., 2025b). RL provides a dynamic alternative by allowing models to learn a reasoning policy through exploration and reward (Guo et al., 2025a; Zhang et al., 2025a). However, its direct application to complex visual reasoning remains challenging due to sparse reward signals (Quadros et al., 2025; Cao et al., 2024), where supervision is only provided at the end of long reasoning trajectories (Wang et al., 2025a). These limitations motivate the development of an RL framework tailored for structured visual tasks that alleviates sparse rewards from the outset.

**Spatial Reasoning on Maps.** Spatial reasoning on transit maps (Wu et al., 2020) has traditionally been approached with multi-stage computer vision systems. These methods first employ Optical Character Recognition (OCR) (Memon et al., 2020; Liu et al., 2024) to extract textual information such as station names, and then use specialized image processing and graph algorithms to identify key topological elements like stations (nodes) and the routes connecting them (edges) (Cherry et al., 2006). Pathfinding is subsequently performed on this extracted graph representation of the transit network using classical search algorithms (Noto & Sato, 2000; Deng et al., 2012). While logical, these systems are often brittle, as errors in early stages can propagate and lead to incorrect final paths. MLLMs offer a promising end-to-end alternative, yet benchmarks consistently show they fail at fundamental spatial tasks like judging relative positions and orientation (Xing et al., 2025). These failures are often attributed to architectural limitations in how vision encoders process positional information (Chen et al., 2024b). When applied directly to map-based planning, MLLMs struggle to comprehend environmental constraints and execute the multi-hop reasoning required (Feng et al., 2025b). Our work addresses above concerns by constructing cold-start data for fine-grained understanding and through multi-stage reinforcement learning training.

## 3 REASONMAP-PLUS CONSTRUCTION

In this section, we first introduce the construction pipeline of REASONMAP-PLUS as shown in Figure 1, which builds on REASONMAP (Feng et al., 2025b), and comprises three stages: (1) data collection and preprocessing, (2) design-guided construction of question–answer pairs, and (3) quality control. We report a comprehensive statistical overview of REASONMAP-PLUS in Appendix A.1.

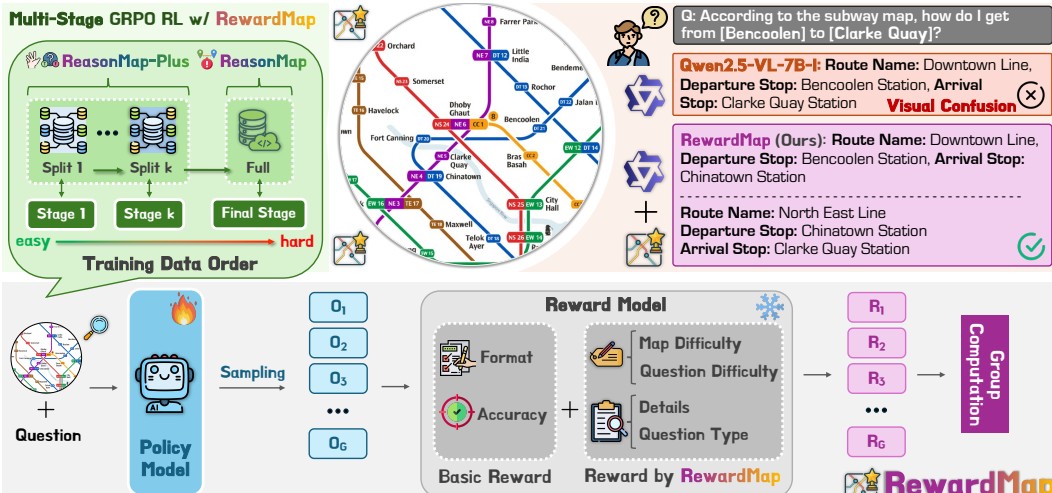

Figure 2: Overview of REWARDMAP. The framework enhances fine-grained visual understanding and reasoning in MLLMs through reinforcement learning with Group Relative Policy Optimization (GRPO). It consists of two key components: (1) a **difficulty-aware reward design** (Section 4.2), which combines format, correctness, and detail rewards with difficulty-based weighting; and (2) a **multi-stage RL curriculum** (Section 4.3), which schedules training data from simple perception tasks to complex reasoning tasks, ensuring effective optimization tackling sparse rewards.

## 3.1 CONSTRUCTION PIPELINE OF REASONMAP-PLUS

We follow the construction pipeline of REASONMAP on stages (1) and (3) for REASONMAP-PLUS. Specifically, for (1) data collection and preprocessing, we reuse the collected high-resolution transit maps and annotated line-stop information (refer to the Metro Data) in REASONMAP to drive the following question-answer generation; for (3) quality control, we manually review the automatically generated question-answer pairs to verify correctness and, when necessary, adjust the question distribution to maintain diversity and a balanced difficulty. We then present the details of stage (2).

**Construction of Question-Answer Pairs in REASONMAP-PLUS.** We extend the planning question in REASONMAP to 5 related categories covering counting and True or False questions (Figure 1), preserving consistency between REASONMAP-PLUS and REASONMAP. For each category, we construct questions based on predesigned question templates (see Appendix A.2) and automatically derive answers from the Metro Data to form question-answer pairs.

We consider the following question types: (1) **Global Counting** assesses global fine-grained visual understanding by asking for the number of lines in a map. This type is sparse, and each map yields only one question; (2) **Local Counting** evaluates local fine-grained visual understanding through two variants: counting the intermediate stops between two specified stops, and counting the number of lines that pass through a specified stop; (3) **True or False** probes fine-grained visual understanding through two variants: judging the spatial relation between two specified stops, and between one stop and one line. We balance yes and no answers to prevent models from exploiting label frequency.

**Difficulty Annotation.** For map difficulty, we follow the manual assignment in REASONMAP, which uniformly categorizes all maps into three levels (*e.g.*, easy, middle, and hard). For question difficulty, these questions probe basic visual understanding of transit maps rather than the complex visual reasoning required by the planning questions in REASONMAP. Accordingly, we define question difficulty by the corresponding map difficulty label.

## 4 METHODOLOGY

In this section, we propose REWARDMAP to enhance fine-grained visual understanding and reasoning in MLLMs. We begin by presenting the overview of our target tasks, baseline, and our proposed REWARDMAP. We then introduce the difficulty-aware reward design and illustrate multi-stage reinforcement learning with the Group Relative Policy Optimization (GRPO) process.

### 4.1 OVERVIEW

**Target Tasks.** We investigate two target tasks in this paper: (1) fine-grained visual understanding in REASONMAP-PLUS (Section 3) and (2) fine-grained visual reasoning in REASONMAP (Feng et al., 2025b). Both are cast as Visual Question Answering (VQA), given a high-resolution image $I$ and an instructional question $Q$, the model must produce an answer $A_{\text{formatted}}$ that conforms to the required output format and correctly addresses $Q$. Unlike conventional benchmarks, our tasks foreground fine-grained perception to assess a model's ability to exploit high-resolution details. Additionally, the route planning task in REASONMAP further evaluates spatial reasoning.

**Baseline.** We conduct baseline experiments under a standard setup with training data from the original REASONMAP. We train the model with GRPO reinforcement learning with a basic reward function consisting of a format reward and a correctness reward. However, training on this setting exhibits sparse rewards under high task difficulty (see the results of Qwen2.5-VL-3/7B-Instruct in Table 1). In GRPO, given an input $x$ and a group $G = \{y_i\}_{i=1}^K$ of sampled outputs with returns $\{r_i\}_{i=1}^K$, the centered group advantage drives policy updates:

$$\hat{A}_i \;=\; r_i \;-\; \frac{1}{K}\sum_{j=1}^{K} r_j, \qquad \max_{\theta}\; \mathcal{L}(\theta) \;=\; \sum_{i=1}^{K} \hat{A}_i \log \pi_\theta(y_i \mid x).$$

With sparse rewards, most $r_i \approx 0$, so $\hat{A}_i$ either collapses to near zero (all failures) or becomes highly skewed (rare positives), yielding low-signal or high-variance gradients and thus slowing convergence.

**REWARDMAP.** To mitigate the sparse reward issue observed in the baseline, we propose RE-WARDMAP, which consists of two components: (1) a detail-oriented reward with difficulty-aware weighting, and (2) a multi-stage RL regimen that exploits dense-reward questions from REASONMAP-PLUS for effective cold start. Implementation details are provided below.

### 4.2 DIFFICULTY-AWARE REWARD DESIGN

Our reward function comprises three terms: format reward, correctness reward, and detail reward, scaled by a difficulty factor and a weighting coefficient for the detail reward:

$$R \;=\; W_{\text{difficulty}}\big(R_{\text{format}} \;+\; R_{\text{correctness}} \;+\; \alpha \times R_{\text{detail}}\big),$$

where $R_{\text{format}}$, $R_{\text{correctness}}$, and $R_{\text{detail}}$ denote the format, correctness, and detail rewards, respectively; $\alpha > 0$ controls the relative strength of the detail term ($\alpha > 0$ is set to $0.5$ for subsequent training); and $W_{\text{difficulty}} > 0$ scales the overall reward according to difficulty.

**Format Reward.** The format reward enforces compliance with task-specific output conventions: for REASONMAP-PLUS, answers are elicited within a "\boxed{}" to localize the output; for REASONMAP, the reward is computed using the benchmark's original formatting specification[1].

**Correctness Reward.** For training data in REASONMAP-PLUS, we use exact-match scoring to compute rewards, as these questions have a single deterministic ground truth (*e.g.*, numerals or yes/no). For training data in our proposed REASONMAP, the correctness reward is computed using the benchmark's official evaluation algorithm[2].

**Detail Reward.** To alleviate sparse rewards caused by the difficulty of planning tasks, we add a detail reward that grants partial credit for correct items of the answer. Specifically, we reward/penalize correctness of the origin and destination stops, route names, transfer stations, and the number of route segments (see the computation pipeline in Algorithm 1). Additionally, we modulate this influence of detail reward on training with a weighting coefficient $\alpha$.

**Difficulty-Aware Weighting.** To incorporate problem difficulty, we scale the sum of the three rewards by a difficulty-aware weight. We consider two fine-grained factors: (1) Map difficulty (for all training data from REASONMAP and REASONMAP-PLUS), with weights assigned by the three levels (*e.g.*, easy, medium, hard, see Appendix A.1); and (2) Question difficulty (for the training data in REASONMAP), with weights determined by the transfer count of the required route. The weighting

---

[1] Please refer to Appendix A.1 in REASONMAP paper for more details.
[2] See Appendix B.1 of the REASONMAP paper (Feng et al., 2025b).

Table 1: Evaluations of reference models and fine-tuned models on REASONMAP and REASONMAP-PLUS. "$S.$" represents results for short questions, while "$L.$" denotes results for long questions. **Bold** indicates the best results among fine-tuned models, while underline represents the second best.

| Model | Training Data | REASONMAP ($S./L.$) | | REASONMAP-PLUS | |
| --- | --- | --- | --- | --- | --- |
| | | Weighted Acc. | Weighted Map Score | Weighted Acc. | Weighted Acc. (Count / TorF) |
| *Reference Models* | | | | | |
| Kimi-VL-A3B-Thinking | - | 5.47% / 5.47% | 2.44 / 3.17 | 33.95% | 18.17% / 50.39% |
| Kimi-VL-A3B-Instruct | - | 12.76% / 12.33% | 3.30 / 5.37 | 32.55% | 14.75% / 51.08% |
| Qwen2.5-VL-3B-Instruct | - | 8.68% / 7.99% | 2.75 / 3.70 | 37.61% | 22.68% / 53.16% |
| Qwen2.5-VL-32B-Instruct | - | 16.49% / 15.71% | 3.88 / 6.84 | 58.32% | 46.96% / 70.14% |
| Qwen2.5-VL-72B-Instruct | - | 26.65% / 24.22% | 5.09 / 8.80 | 53.21% | 43.46% / 63.36% |
| Seed1.5-VL | - | 34.20% / 38.02% | 5.25 / 11.96 | 73.58% | 65.26% / 82.23% |
| GPT-4o | - | 41.15% / 42.80% | 6.84 / 13.57 | 64.42% | 59.28% / 69.77% |
| GPT-5 | - | 59.98% / 62.50% | 9.48 / 19.75 | 88.95% | 86.40% / 91.60% |
| *Baseline &* REWARDMAP | | | | | |
| Qwen2.5-VL-7B-Instruct | - | 13.28% / 7.12% | 4.01 / 5.74 | 44.21% | 37.39% / 51.32% |
| + RL (GRPO) | $R_{train}$ | 26.22% / 26.04% | 5.52 / 9.52 | 44.64% | 37.57% / 52.01% |
| + RL (REINFORCE++) | $R_{train}$ | 27.17% / 27.60% | 5.68 / 10.12 | 44.64% | 36.82% / 52.79% |
| + RL (ReMax) | $R_{train}$ | 26.22% / 27.26% | 5.57 / 9.99 | 45.39% | 38.37% / 52.70% |
| + SFT | $RPlus_{train}$ | 13.63% / 9.11% | 4.09 / 6.25 | 57.93% | 50.73% / 65.44% |
| + SFT → RL | $RPlus_{train} + R_{train}$ | 28.82% / 30.38% | 5.88 / 10.62 | 60.53% | 55.38% / 65.90% |
| + RL (baseline) | $RPlus_{train} + R_{train}$ | 29.51% / 29.51% | 6.00 / 10.41 | 67.61% | 68.37% / 66.82% |
| + REWARDMAP | $RPlus_{train} + R_{train}$ | **31.51% / 31.77%** | **6.21 / 11.22** | **74.25%** | **72.18% / 76.42%** |

scheme is defined as follows:

$$W_{\text{difficulty}} = W_{\text{map}} + W_{\text{question}},$$

$$W_{\text{map}} = \begin{cases} \gamma_e, & \text{map difficulty} = \text{easy} \\ \gamma_m, & \text{map difficulty} = \text{medium} \\ \gamma_h, & \text{map difficulty} = \text{hard} \end{cases}, \qquad W_{\text{question}} = \begin{cases} \beta_0, & \text{transfer count} = 0 \\ \beta_1, & \text{transfer count} \geq 1 \end{cases}.$$

### 4.3 MULTI-STAGE GRPO-BASED REINFORCEMENT LEARNING

To effectively exploit REASONMAP-PLUS and alleviate the sparse-reward issue in complex tasks such as route planning in REASONMAP, we design a **multi-stage curriculum** built upon GRPO-based reinforcement learning. The core idea is to progressively schedule training data in a principled manner, ensuring a smoother optimization process and more stable reward propagation. We adhere to two complementary principles:

**(1) Global curriculum principle.** We impose a coarse-to-fine learning schedule by partitioning tasks into distinct stages according to both *question type* (from binary judgment → counting → planning) and *target task* (from fine-grained visual understanding → fine-grained visual reasoning). This global ordering ensures that the agent acquires fundamental perceptual skills before engaging in more abstract reasoning, as illustrated in Figure 2.

**(2) Local stochasticity principle.** Within each stage, we avoid strict deterministic ordering by introducing randomness, *i.e.*, shuffling training samples instead of ranking them solely by heuristic difficulty metrics (*e.g.*, map or question complexity). This stochasticity prevents overfitting to a fixed curriculum trajectory and enhances robustness.

By jointly applying these principles, the proposed multi-stage RL scheme transforms curriculum training into a structured combination of *reward shaping* and *task scheduling*, thereby enabling effective reinforcement learning on inherently sparse-reward visual reasoning problems.

## 5 EXPERIMENTS

### 5.1 EXPERIMENTAL SETUPS

**Training Details.** We conduct training experiments under various settings with the training data from REASONMAP (696 samples, $R_{train}$) (Feng et al., 2025b) and our proposed REASONMAP-PLUS (2,570 samples, $RPlus_{train}$) on Qwen2.5-VL models (Bai et al., 2025b) with 8 NVIDIA H800 GPUs. For GRPO (Shao et al., 2024) RL training, we use AdamW with an initial learning rate of

Table 2: Evaluation of reference models and fine-tuned models on various benchmarks. **Bold** indicates the best results among fine-tuned models, while underline represents the second best. $\dagger, \ddagger, \$, *, \S$ denote the results from the technical report or the official HuggingFace repository (see result sources in Appendix C.3), while all other results are obtained from our own experiments.

| Model | Spatial Reasoning | | Fine-grained Visual Reasoning | | General Task | | Avg. |
|---|---|---|---|---|---|---|---|
| | SEED-Bench-2-Plus | SpatialEval | $V^*$ | HRBench | ChartQA | MMStar | |
| *Reference Models* | | | | | | | |
| Kimi-VL-A3B-Instruct | 58.49% | 52.64% | 59.16% | 55.38% | 87.08% | 61.70%* | 62.41% |
| Qwen2.5-VL-3B-Instruct | 58.86% | 55.04% | 58.12% | 66.25% | 84.00%$^\ddagger$ | 55.90%$^\ddagger$ | 63.03% |
| Qwen2.5-VL-32B-Instruct | 72.10%$^\$$ | 57.99% | 82.72% | 63.50% | 64.90%$^\$$ | 69.50%$^\S$ | 68.45% |
| Qwen2.5-VL-72B-Instruct | 73.00%$^\$$ | - | 86.40%$^\dagger$ | - | 89.50%$^\ddagger$ | 70.80%$^\ddagger$ | - |
| Seed1.5-VL | - | - | 89.00%$^\dagger$ | - | 87.40%$^\dagger$ | 76.20%$^\dagger$ | - |
| *Baseline & REWARDMAP* | | | | | | | |
| Qwen2.5-VL-7B-Instruct | 60.97% | 57.30% | 78.01% | 68.75% | 86.12% | 61.67% | 68.80% |
| + SFT → RL | 61.59%$_{\uparrow 0.62\%}$ | 69.06%$_{\uparrow 11.76\%}$ | 78.01%$_{\uparrow 0.00\%}$ | 71.00%$_{\uparrow 2.25\%}$ | 86.92%$_{\uparrow 0.80\%}$ | **62.27%**$_{\uparrow 0.60\%}$ | 71.48%$_{\uparrow 2.68\%}$ |
| + REWARDMAP | **61.96%**$_{\uparrow 0.99\%}$ | **70.81%**$_{\uparrow 13.51\%}$ | **80.10%**$_{\uparrow 2.09\%}$ | **71.25%**$_{\uparrow 2.50\%}$ | **87.24%**$_{\uparrow 1.12\%}$ | **62.27%**$_{\uparrow 0.60\%}$ | **72.27%**$_{\uparrow 3.47\%}$ |

$1.0 \times 10^{-6}$ and a KL divergence coefficient of $1.0 \times 10^{-3}$. The global batch size is 16, and we sample 8 responses per query. Besides RL training, we conduct baseline experiments using SFT with training data from REASONMAP-PLUS. We apply LoRA (Hu et al., 2022) to the language blocks and the text-vision projector with an initial learning rate of $1.0 \times 10^{-4}$. For implementation, we adopt LLaMA-Factory (Zheng et al., 2024) and VeRL (Sheng et al., 2024). Additionally, we include two RL training baselines with REINFORCE++ (Hu et al., 2025a) and ReMax (Li et al., 2023).

**Inference Details.** All evaluations use greedy decoding (temperature=0). For open-source models, we cap the maximum output length at $2,048$ tokens and retain all other settings from the official HuggingFace configurations; deployments use PyTorch with Transformers[3] on 8 NVIDIA H800 GPUs. For closed-source models, we evaluate through official APIs using default settings. The evaluated models include: Kimi-VL-A3B-Insturct/Thinking (Team et al., 2025), Qwen2.5-VL-3/7/32/72B-Instruct (Bai et al., 2025b), Seed1.5-VL (Guo et al., 2025b), GPT-5/4o (OpenAI, 2024).

**Evaluation Datasets.** We first evaluate on the test sets of REASONMAP and REASONMAP-PLUS, and we report the metrics adjusted by the difficulty-aware weighting scheme (see details in Appendix C.1). To further assess the capability gains brought by REWARDMAP, we next employ six widely-used benchmarks spanning three dimensions (*e.g.*, spatial reasoning, fine-grained visual reasoning, and general tasks): SEED-Bench-2-Plus (Li et al., 2024), SpatialEval (Wang et al., 2024a), $V^*$Bench Wu & Xie (2024), HRBench (Wang et al., 2025f), ChartQA (Masry et al., 2022), and MMStar (Chen et al., 2024a) (see Appendix C.2 for details). All evaluations are conducted with VLMEvalKit[4].

## 5.2 MAIN RESULTS

**Results on REASONMAP and REASONMAP-PLUS.** We evaluate the proposed REWARDMAP on REASONMAP and REASONMAP-PLUS, both of which provide fine-grained difficulty annotations to assess visual understanding, visual reasoning, and spatial reasoning. In addition to the baselines introduced in Section 4.1, we compare against two widely-used settings: (1) **SFT → RL baseline**, which applies SFT on REASONMAP-PLUS followed by RL on REASONMAP, and (2) **RL baseline**, which performs RL on the combined training data from REASONMAP and REASONMAP-PLUS. For both baselines, the reward design includes only format and correctness terms.

As shown in Table 1, REWARDMAP consistently outperforms all baselines across different question templates in REASONMAP and question types of REASONMAP-PLUS. On REASONMAP, it substantially surpasses the best open-source result (Qwen2.5-VL-72B-Instruct) and approaches the performance of the closed-source model (Seed1.5-VL). On REASONMAP-PLUS, REWARDMAP not only exceeds all open-source models but also outperforms Seed1.5-VL.

**Results on Other Benchmarks.** We further evaluate the generalization ability of our method on six benchmarks spanning three task categories introduced in Section 5.1. Table 2 reports the results of reference models, the SFT → RL baseline, and our proposed REWARDMAP. Across all six benchmarks, REWARDMAP achieves consistent improvements, with the most substantial gain of

---

[3]https://github.com/huggingface/transformers
[4]https://github.com/open-compass/VLMEvalKit

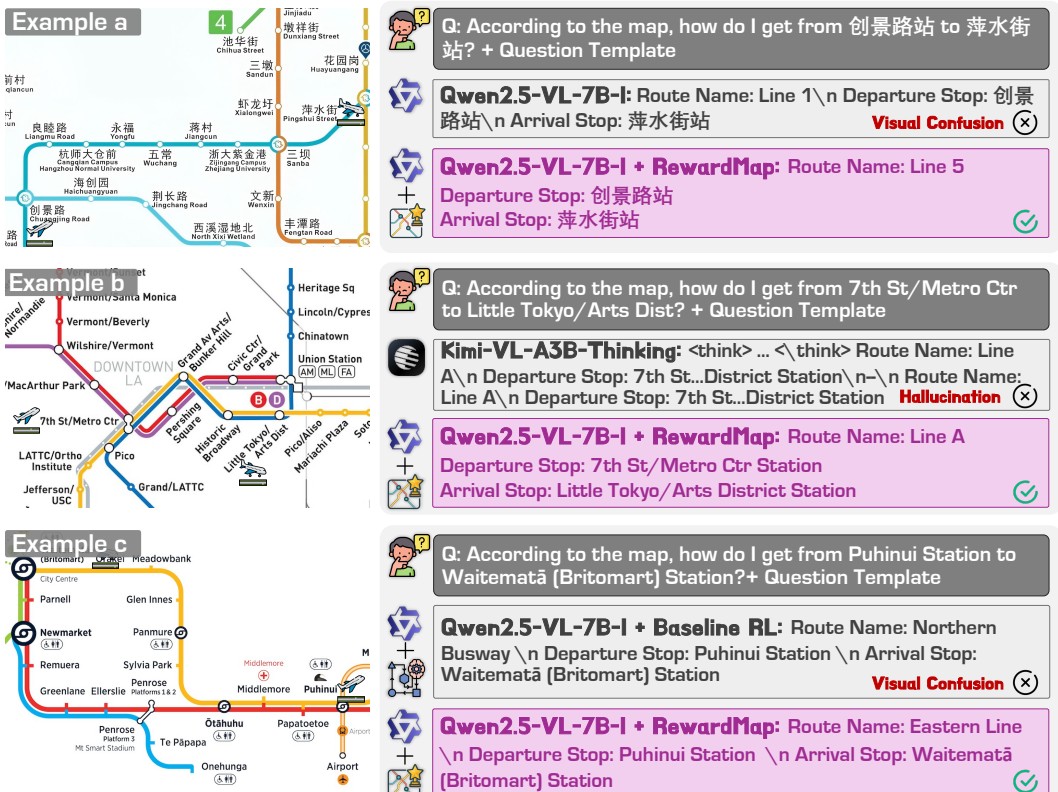

Figure 3: Qualitative comparisons among reference models, baseline, and our proposed Re-WARDMAP. We crop and zoom in on the transit map for clearer presentation.

13.51% observed on SpatialEval. While the SFT → RL baseline also yields stable improvements, its performance remains inferior to REWARDMAP. These results highlight the significant contributions of both REWARDMAP and the datasets (both REASONMAP and REASONMAP-PLUS) to enhancing model general capability. Beyond the aggregate results, we further report task-wise improvements. On SEED-Bench-2-Plus, REWARDMAP improves the map split. For SpatialEval, accuracy rises substantially on *mazenav* (19.60% → 57.20%) and moderately on *spatialreal* (70.37% → 72.59%). On HRBench, performance increases for both single-image reasoning (85.25% → 88.00%) and cross-image alignment (52.25% → 54.00%). For MMStar, REWARDMAP achieves steady gains across fine-grained perception (59.60% → 60.80%), instance reasoning (68.00% → 71.20%), and mathematical reasoning (59.60% → 63.20%). These results show that REWARDMAP consistently strengthens visual grounding and robustness in diverse spatial and reasoning tasks.

## 5.3 QUALITATIVE RESULTS

Figure 3 presents a qualitative comparison between reference models, the baseline RL model, and our proposed REWARDMAP. Across diverse maps, we observe that reference models and the baseline RL model often suffer from visual confusion (*e.g.*, mistaking the route or stop as shown by Example a & c in Figure 3) or even hallucination (*e.g.*, repeating the same route many times in Figure 3 (b)). In contrast, REWARDMAP consistently produces the correct target routes. These examples highlight the effectiveness of REWARDMAP in handling visually complex maps (*e.g.*, Example a & b in Figure 3) and reducing both visual confusions and hallucinations. Additionally, we present further comparison cases that provide additional evidence that REWARDMAP improves visual grounding and reduces the likelihood of visual confusion or hallucination. These examples are included in Appendix D.1.

## 5.4 DIAGNOSTIC EXPERIMENTS

We provide extensive ablation studies using Qwen2.5-VL-7B-Instruct, and further verify the effectiveness of REWARDMAP in addressing sparse-reward issues, across different model scales (*e.g.*, Qwen2.5-VL-3B-Instruct) and different model architectures (*e.g.*, Kimi-VL-A3B-Instruct).

Table 3: Ablation on reward design and multi-stage design of REWARDMAP. "*S.*" represents results for short questions, while "*L.*" denotes results for long questions.

| Model | Training Data | REASONMAP (S./L.) | | REASONMAP-PLUS | |
| --- | --- | --- | --- | --- | --- |
| | | Weighted Acc. | Weighted Map Score | Weighted Acc. | Weighted Acc. (Count / TorF) |
| RL (baseline) | $R_{train}$ | 26.22% / 26.04% | 5.52 / 9.52 | 44.64% | 37.57% / 52.01% |
| RL + Reward Design | $R_{train}$ | 29.08% / 29.95% | 5.88 / 10.53 | 45.16% | 37.79% / 52.84% |
| RL (baseline) | $RPlus_{train} + R_{train}$ | 29.51% / 29.51% | 6.00 / 10.41 | 67.61% | 68.37% / 66.82% |
| RL + Reward Design | $RPlus_{train} + R_{train}$ | 30.56% / 30.38% | 6.12 / 10.62 | 71.07% | 67.61% / 74.67% |
| RL + Multi-Stage Design | $RPlus_{train} + R_{train}$ | 30.64% / 31.51% | 6.08 / 10.88 | 73.12% | 69.52% / 76.88% |
| REWARDMAP | $RPlus_{train} + R_{train}$ | **31.51% / 31.77%** | **6.21 / 11.22** | **74.25%** | **72.18% / 76.42%** |

Table 4: Ablation on REASONMAP-PLUS. "*S.*" represents results for short questions, while "*L.*" denotes results for long questions.

| Model | Training Data | REASONMAP (S./L.) | | REASONMAP-PLUS | |
| --- | --- | --- | --- | --- | --- |
| | | Weighted Acc. | Weighted Map Score | Weighted Acc. | Weighted Acc. (Count / TorF) |
| RL (baseline) | $R_{train}$ | 26.22% / 26.04% | 5.52 / 9.52 | 44.64% | 37.57% / 52.01% |
| RL (baseline) | $RPlus_{train} + R_{train}$ | 29.51% / 29.51% | 6.00 / 10.41 | 67.61% | 68.37% / 66.82% |
| SFT → RL | $RPlus_{train} + R_{train}$ | 28.82% / 30.38% | 5.88 / 10.62 | 60.53% | 55.38% / 65.90% |

**Ablation on Reward Design and Multi-Stage Design of REWARDMAP.** We ablate the reward design and multi-stage design of REWARDMAP under two training data settings. For the configuration using only REASONMAP training data, where the multi-stage design cannot be applied, we ablate the reward design alone. As shown in Table 3, enabling either component individually yields performance gains on both REASONMAP and REASONMAP-PLUS, while combining them achieves the best results, confirming the effectiveness and complementarity of the two components.

**Ablation on REASONMAP-PLUS.** Next, we assess the impact of REASONMAP-PLUS under two training strategies (*e.g.*, SFT and RL). As shown in Table 4, conducting cold-start training with REASONMAP-PLUS consistently improves performance on both REASONMAP and REASONMAP-PLUS, regardless of the chosen strategy.

**Ablation on Granularity of Multi-Stage Design.** We further conduct an ablation study on the granularity of the multi-stage design by comparing a coarse-grained variant with our proposed REWARDMAP. As shown in Table 5, switching to the coarse strategy leads to performance degradation while still outperforming the baseline, thereby reinforcing the effectiveness of the multi-stage design.

**Ablation on the Hyperparameter $\alpha$.** Based on the full REWARDMAP pipeline, we varied only $\alpha$. The results are reported in Table 6. We observe a slight performance drop when $\alpha$ is small, while the performance becomes similar once $\alpha$ enters a higher range. These results suggest that the detailed reward is indeed beneficial (as $\alpha$ determines its contribution to the total reward), and they also indicate that $\alpha$ is not sensitive within a reasonable interval.

**Ablation on the Difficulty-Aware Weighting Scheme.** Using the REASONMAP training set, we further conduct two ablations: one varying only the relative magnitudes of $\gamma_{e/m/h}$ and another varying only those of $\beta_{0/1}$. As shown in Table 7, performance decreases slightly when the weights become nearly uniform, but remains stable once clear difficulty distinctions are introduced. These results support the effectiveness of our difficulty-aware weighting—where $\gamma_{e/m/h}$ and $\beta_{0/1}$ modulate map and question difficulty, respectively—and indicate that both weights are insensitive within a reasonable range.

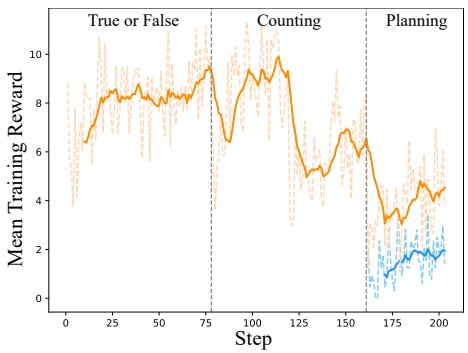

Figure 4: Comparison of training rewards between baseline RL and REWARDMAP. The yellow curve denotes the reward trajectory of REWARDMAP, while the blue curve corresponds to the baseline RL trained solely on REASONMAP.

**Effectiveness of Tackling Sparse Reward.** We validate the effectiveness of REWARDMAP to address sparse rewards. As shown in Figure 4, we compare reward trajectories of REWARDMAP with a baseline RL trained on REASONMAP, aligned at the planning stage. The results show that REWARDMAP alleviates reward sparsity, further confirming its effectiveness.

Table 5: Ablation on granularity of multi-stage design. "$S$." represents results for short questions, while "$L$." denotes results for long questions.

| Model | Training Data | ReasonMap ($S./L.$) | | ReasonMap-Plus | |
| --- | --- | --- | --- | --- | --- |
| | | Weighted Acc. | Weighted Map Score | Weighted Acc. | Weighted Acc. (Count / TorF) |
| RL (baseline) | $RPlus_{train} + R_{train}$ | 29.51% / 29.51% | 6.00 / 10.41 | 67.61% | 68.37% / 66.82% |
| RL (coarse-grained Multi-Stage) | $RPlus_{train} + R_{train}$ | 29.60% / 30.21% | 5.98 / 10.83 | 70.30% | 66.02% / 74.76% |
| RL (Multi-Stage) = RewardMap | $RPlus_{train} + R_{train}$ | 31.51% / 31.77% | 6.21 / 11.22 | 74.25% | 72.18% / 76.42% |

Table 6: Ablation on the hyperparameter $\alpha$. "$S$." represents results for short questions, while "$L$." denotes results for long questions.

| Model | Training Data | ReasonMap ($S./L.$) | | ReasonMap-Plus | |
| --- | --- | --- | --- | --- | --- |
| | | Weighted Acc. | Weighted Map Score | Weighted Acc. | Weighted Acc. (Count / TorF) |
| + RewardMap [$\alpha$=0.3] | $RPlus_{train} + R_{train}$ | 30.73% / 31.16% | 6.01 / 10.81 | 72.08% | 68.41% / 75.91% |
| + RewardMap [$\alpha$=0.5] | $RPlus_{train} + R_{train}$ | 31.51% / 31.77% | 6.21 / 11.22 | 74.25% | 72.18% / 76.42% |
| + RewardMap [$\alpha$=0.7] | $RPlus_{train} + R_{train}$ | 32.03% / 32.20% | 6.23 / 11.20 | 72.81% | 70.14% / 75.59% |

Table 7: Ablation on the difficulty-aware weights. "$S$." represents results for short questions, while "$L$." denotes results for long questions.

| Model | Training Data | ReasonMap ($S./L.$) | | ReasonMap-Plus | |
| --- | --- | --- | --- | --- | --- |
| | | Weighted Acc. | Weighted Map Score | Weighted Acc. | Weighted Acc. (Count / TorF) |
| Qwen2.5-VL-7B-Instruct | - | 13.28% / 7.12% | 4.01 / 5.74 | 44.21% | 37.39% / 51.32% |
| + Reward Design [$\gamma_{e/m/h} = (1.0, 1.1, 1.2)]R_{train}$ | | 26.91% / 28.47% | 5.64 / 10.24 | 45.34% | 37.97% / 53.02% |
| + Reward Design [$\gamma_{e/m/h} = (1.0, 1.2, 1.5)]R_{train}$ | | 29.08% / 29.95% | 5.88 / 10.53 | 45.16% | 37.79% / 52.84% |
| + Reward Design [$\gamma_{e/m/h} = (1.0, 1.5, 2.0)]R_{train}$ | | 29.86% / 28.99% | 5.93 / 10.31 | 45.16% | 37.31% / 53.35% |
| + Reward Design [$\beta_{0/1} = (0.0, 0.2)$] | $R_{train}$ | 28.30% / 29.34% | 5.68 / 10.12 | 44.60% | 37.39% / 52.10% |
| + Reward Design [$\beta_{0/1} = (0.0, 0.5)$] | $R_{train}$ | 29.08% / 29.95% | 5.88 / 10.53 | 45.16% | 37.79% / 52.84% |
| + Reward Design [$\beta_{0/1} = (0.0, 0.8)$] | $R_{train}$ | 29.60% / 29.34% | 5.89 / 10.23 | 45.39% | 38.68% / 52.38% |

Table 8: Evaluation of RewardMap across model scales. "$S$." represents results for short questions, while "$L$." denotes results for long questions.

| Model | Training Data | ReasonMap ($S./L.$) | | ReasonMap-Plus | |
| --- | --- | --- | --- | --- | --- |
| | | Weighted Acc. | Weighted Map Score | Weighted Acc. | Weighted Acc. (Count / TorF) |
| Qwen2.5-VL-3B-Instruct | - | 8.68% / 7.99% | 2.75 / 3.70 | 37.61% | 22.68% / 53.16% |
| + RL (baseline) | $R_{train}$ | 11.46% / 10.50% | 3.81 / 6.09 | 38.29% | 22.06% / 55.19% |
| + RewardMap | $RPlus_{train} + R_{train}$ | **19.36% / 15.89%** | **4.79 / 7.53** | **65.91%** | **63.58% / 68.34%** |

**Effectiveness of RewardMap across Model Scales.** We evaluate the effectiveness of RewardMap across different model scales. Due to training cost constraints, we adopt Qwen2.5-VL-3B-Instruct as the base model and compare the baseline RL with RewardMap. As shown in Table 8, RewardMap achieves the promising results, demonstrating its robustness and effectiveness.

Table 9: Results on Kimi-VL (Kimi-VL-A3B-Instruct). "$S$." represents results for short questions, while "$L$." denotes results for long questions.

| Model | Training Data | ReasonMap ($S./L.$) | | ReasonMap-Plus | |
| --- | --- | --- | --- | --- | --- |
| | | Weighted Acc. | Weighted Map Score | Weighted Acc. | Weighted Acc. (Count / TorF) |
| Kimi-VL-A3B-Instruct | - | 12.76% / 12.33% | 3.30 / 5.37 | 32.55% | 14.75% / 51.08% |
| + RewardMap | $R_{train}$ | 18.58% / 17.36% | 4.70 / 7.69 | 35.92% | 15.20% / 57.50% |

**Generalization of RewardMap across Model Architectures.** We further trained Kimi-VL (Kimi-VL-A3B-Instruct) with the RewardMap pipeline using the training set from ReasonMap. As shown in Table 9, the model achieves substantial performance gains, demonstrating that our method generalizes beyond the Qwen2.5-VL series.

## 6 CONCLUSION

In this work, we address the challenge of applying reinforcement learning to fine-grained visual reasoning, where sparse rewards and long reasoning horizons often hinder effective optimization. Building upon the ReasonMap benchmark, we introduce ReasonMap-Plus, an extended dataset that organizes tasks along a difficulty continuum, providing dense supervision to facilitate cold-start training. Furthermore, we propose RewardMap, a multi-stage reinforcement learning framework that combines curriculum-style task scheduling with difficulty-aware reward design. Our experiments demonstrate that each of these components contributes to stable and effective training, and that their integration yields the strongest improvements. RewardMap not only advances performance on ReasonMap and ReasonMap-Plus but also enhances robustness across broader visual reasoning benchmarks, indicating improved perceptual and reasoning capabilities of multimodal models.

## ACKNOWLEDGEMENT

This paper is supported by Young Scientists Fund of the National Natural Science Foundation of China (NSFC) (No. 62506305), Zhejiang Leading Innovative and Entrepreneur Team Introduction Program (No. 2024R01007), Key Research and Development Program of Zhejiang Province (No. 2025C01026), Scientific Research Project of Westlake University (No. WU2025WF003), Chinese Association for Artificial Intelligence (CAAI) & Ant Group Research Fund - AGI Track (No. 2025CAAI-ANT-13). It is also supported by the research funds of the National Talent Program and Hangzhou Municipal Talent Program.

## ETHICS STATEMENT

This work does not involve human subjects or sensitive personal data. REASONMAP-PLUS is constructed from publicly available transit maps with automatically generated question–answer pairs, ensuring no privacy or security concerns. The datasets are released exclusively for academic research under the Apache License 2.0 on HuggingFace, and all reported results are fully reproducible with the released code and configurations.

## REPRODUCIBILITY STATEMENT

To ensure reproducibility, we provide the evaluation setup details in Section 5.1 and Appendix C, including hardware and implementation, which facilitates rapid replication.

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

APPENDIX

In Appendix A, we provide a statistical overview of REASONMAP-PLUS, question templates, and fine-grained annotation. Appendix B outlines the computation pipeline of the detail reward. Appendix C describes evaluation settings, including difficulty-aware weighting, benchmark datasets, and result sources in Table 2. In Appendix D, we provide more cases to support our claims in Section 5.3. We discuss future work in Appendix E. Finally, Appendix F presents the statement on LLM usage.

A  DATASET CONSTRUCTION DETAILS

A.1  STATISTICAL OVERVIEW

REASONMAP-PLUS mirrors REASONMAP in image composition, comprising high-resolution transit maps from 30 cities. It contains $4,018$ questions across five categories (*e.g.*, Local Counting 1 - $1,050$; Local Counting 2 - 970; Global Counting - 30; True or False 1 - $1,039$; True or False 2 - 929), with a difficulty distribution of $1,259$ easy, $1,342$ middle, and $1,417$ hard. To preserve difficulty balance and diversity, we adopt the same split as REASONMAP, using questions from 11 cities for the test set ($1,448$) and the remainder for the training set ($2,570$).

A.2  QUESTION TEMPLATE

We present the question templates of REASONMAP-PLUS as follows.

> **Local Counting 1**
>
> Please solve the multiple choice problem and put your answer (one of ABCD) in one "
> boxed". According to the subway map, how many intermediate stops are there between stop
> 1 and stop 2 (except for this two stops)? A) x B) x C) x D) x

**Local Counting 2**

Please solve the problem and put your answer in one "
boxed". According to the subway map, how many lines pass through stop 1?

**Global Counting**

Please solve the problem and put your answer in one "
boxed". According to the subway map, how many subway (metro) lines are there in total?

**True or False 1**

Please solve the problem and put your answer (only answer yes or no) in one "
boxed". According to the subway map, is it true that stop 1 is the same line as stop 2?

**True or False 2**

Please solve the problem and put your answer (only answer yes or no) in one "
boxed". According to the subway map, is it true that stop 1 is on the line x?

## A.3 FINE-GRAINED ANNOTATION

We provide more details of the process-level annotation in this section. The construction involves two steps. First, we annotate the route–stop structure of each transit map, including the ordered stops for every line, interchange stations, and branching points. These annotations, as listed in the Meta Data, are recorded in a JSON file. An excerpt for Singapore is shown below:

**Meta Data**

"Circle Line": ["HarbourFront (Transfer)", "Telok Blangah", "Labrador Park", ...],
"Downtown Line": [...],
...

Based on the Meta Data, we generate process-level annotations with task-specific Python scripts. For example, in the Local Counting 1 category of REASONMAP-PLUS, where the question asks for the number of intermediate stops between two stations, the script identifies the two queried stops, extracts all intermediate stops between them, computes the count as the final answer, and records the intermediate stops as the process-level annotation. Other task types, such as True/False or shortest-path queries, follow a similar procedure.

---

**Example 1 (Local Counting 1)**

"country": "singapore",
"city": "singapore",
"station_1": "Clarke Quay",
"station_2": "Farrer Park",
"figure": "./maps/singapore/singapore.png",
"question": "According to the map, how many intermediate stops are there
between Clarke Quay and Farrer Park (excluding the endpoints)?",
"answer": "2",
"fine-grained answer": ["Dhoby Ghaut", "Little India"],
"type": "counting_1",
"difficulty_city": "hard",
"city_line_count": "6",
"city_transfer_count": "53",
"json": "./stations/singapore/singapore.json"

---

**Example 2 (True or False 1)**

"country": "singapore",
"city": "singapore",
"station_1": "Somerset",
"station_2": "Bras Basah",
"figure": "./maps/singapore/singapore.png",
"question": "According to the map, is it true that Somerset is
on the same line as Bras Basah?",
"answer": "No",
"fine-grained answer":  "Somerset": "North South Line", "Bras Basah": "Circle Line" ,
"type": "torf_1",
"difficulty_city": "hard",
"city_line_count": "6",
"city_transfer_count": "53",
"json": "./stations/singapore/singapore.json"

---

## B    REWARDMAP DETAILS

### B.1    COMPUTATION PIPELINE OF DETAIL REWARD

We present the complete computation pipeline of Detail Reward in Algorithm 1.

## C    EVALUATION DETAILS

### C.1    DETAILS OF DIFFICULTY-AWARE WEIGHTING

For REASONMAP, we follow the weighting scheme described in the original paper (see Appendix B.3 therein). For REASONMAP-PLUS, weights are assigned solely based on map difficulty, with values of 1.0, 1.5, and 2.0 for easy, medium, and hard maps, respectively.

### C.2    DETAILS OF EVALUATION DATASETS

We provide a brief description of the six evaluation benchmarks used in our paper:

1. **SEED-Bench-2-Plus (Map)** (Li et al., 2024) describes three categories (Charts, Maps, Webs) with human-verified multiple-choice items, from which we use the Map slice.

---

**Algorithm 1:** Detailed Reward for Planning Questions in REASONMAP

---

Initialize `score` ← 0;
**if** `route_data` *is empty* **or** *format is wrong* **then**
  **return** `score`;

**if** *departure stop of first segment* = *stop_1* **or** *arrival stop of last segment* = *stop_2* **then**
  `score` ← `score` +2;

**foreach** *segment* $s_i$ *in predicted route* **do**
  **if** `current_transfer_times` > `question_transfer_count` **then**
    `score` ← `score` −5;
  **if** `current_transfer_times` = 0 **and** `is_correct(route name)` **then**
    `score` ← `score` +4;
  **if** *departure stop* ∈ `stations` **and** *arrival stop* ∈ `stations` **then**
    **if** *segment* $s_i$ *is not the last* **then**
      **if** *arrival stop* = *departure stop of next segment* **then**
        `score` ← `score` +1;

`score` ← min(`score`, 10);
**return** `score`;

---

2. **SpatialEval** (Wang et al., 2024a) targets spatial intelligence across relationships, position, counting, and navigation.

3. $V^*$**Bench** (Wu & Xie, 2024) evaluates fine-grained attribute recognition and spatial relationships on high-resolution images.

4. **HRBench** (Wang et al., 2025f) evaluates MLLMs on 4K/8K high-resolution images and introduces a training-free enhancement baseline.

5. **ChartQA** (Masry et al., 2022) benchmarks QA over charts with visual and logical reasoning.

6. **MMStar** (Chen et al., 2024a) offers 1,500 human-curated, vision-indispensable samples covering 6 core capabilities and 18 axes.

### C.3 RESULT SOURCE SUMMARY

We present sources of results in Table 2: †, ‡, $, ∗, § correspond to the Seed1.5-VL technical report (Guo et al., 2025b), the Qwen2.5-VL technical report (Bai et al., 2025b), the InternVL3.5 technical report (Wang et al., 2025e), the Kimi-VL technical report Team et al. (2025), and the official Hugging Face repository of Qwen2.5-VL-32B-Instruct[5], respectively.

## D SUPPLEMENTARY RESULTS

### D.1 COMPARISON CASES

We provide more comparison cases to support deeper analysis in Section 5.3. In the following case, the prediction from Qwen2.5-VL-7B-Instruct exhibits a visual confusion error at *<wrong1>* and a hallucination at *<wrong2>*, misidentifying three intermediate stops but selecting option A (6). In contrast, the RewardMap-trained model correctly identifies a single intermediate stop and selects the correct option C.

> **Question:** *How many intermediate stops are there between A Station and B Station?*
> (`<budapest.png>` + Format Requirement + {A: 6, B: 8, C: 1, D: 9})

---

[5] https://huggingface.co/Qwen/Qwen2.5-VL-32B-Instruct

**Answer (Qwen2.5-VL-7B-Instruct, Wrong).**
To determine the number of intermediate stops between A Station and B Station, we need to look at the yellow line on the map, which represents Metro Line 1.
1. Identify the stations on the yellow line between A and B. 2. Count the number of stations, excluding A and B.
From the map: – A
– Széchenyi Thermal Bath
– Bajza Street
– B
There are 3 intermediate stops between A and B. *<wrong1>*
Therefore, the correct answer is: $\boxed{A}$ *<wrong2>*

**Answer (Qwen2.5-VL-7B-Instruct + RewardMap, Correct).**
To solve this problem, we need to count the intermediate stops between A Station and B Station, excluding these two stops.
Starting from A: – The next stop is Széchenyi Thermal Bath.
Thus, the only intermediate stop between A and B is Széchenyi Thermal Bath.
Therefore, the correct answer is: $\boxed{C}$

## E    FUTURE WORK

Our framework is not limited to transit maps and can be extended to other structured visual domains such as charts and diagrams. Our results support this direction: ChartQA (Masry et al., 2022) improves accuracy from $86.12\% \rightarrow 87.24\%$, Charxiv (Wang et al., 2024b) from $66.41\% \rightarrow 68.90\%$, and InfoVQA (Mathew et al., 2022) from $82.22\% \rightarrow 82.71\%$. These gains suggest that the detail rewards and multi-stage RL scheme generalize to structurally similar tasks. Future work will explore domain-agnostic rewards and broader applications across diverse structured visual reasoning settings.

Efficient methods can further improve the fine-tuned model by our framework (Feng et al., 2024; Bai et al., 2025a; Zhu et al., 2025; Shao et al., 2025a;b; Tao et al., 2025a;b;c). The idea of detail reward can be extended to other fields that require reasoning capability (Zhang et al., 2025d; 2026; Wang et al., 2025d;c; Ai et al., 2025; Wu et al., 2025b; Li et al., 2025e; Wang et al., 2023; Jin et al., 2025; Xiao et al., 2025; Li et al., 2025a; Feng et al., 2026; Wang et al., 2026). Leveraging reinforcement learning for safety alignment (Zhang et al., 2025b; Li & Wang, 2026; Li et al., 2025b) remains a promising direction for further exploration.

## F    LARGE LANGUAGE MODEL USAGE STATEMENT

Large Language Models (LLMs) were used only for surface-level editing of the manuscript (e.g., polishing grammar and style, rephrasing for clarity, and making minor LaTeX adjustments). They were not involved in generating ideas, methods, algorithms, code, experiments, figures, tables, or citations. All research design, implementation, data processing, and analysis were performed by the authors. LLM use was limited to de-identified text snippets, with no proprietary data or unpublished results shared, and all outputs were manually reviewed and revised. This limited assistance does not affect reproducibility, as every reported result is fully reproducible from the released code and configurations.

