# OpenReview forum: "RewardMap: Tackling Sparse Rewards in Fine-grained Visual Reasoning via Multi-Stage Reinforcement Learning"
_ICLR.cc/2026/Conference — ICLR 2026 Poster_

### Official Review · Reviewer_ekjh · 2025-10-18

**Soundness:** 3
**Presentation:** 2
**Contribution:** 1
**Rating:** 2
**Confidence:** 4

**Summary:**

This paper tackles the problem of improving reasoning of multi-modal large language models. They use the REASONMAP dataset and show that directly performing GRPO on the dataset is not very effective for multi-modal reasoning because of high difficulty of questions and sparse rewards. To alleviate this, they construct REASONMAP-PLUS, an augmented dataset which easier VQA style queries with denser reward. They also show improvements in training MLLMs using GRPO by adding a curriculum (easy->hard) questions and by adding partial rewards to alleviate the problem of sparse rewards. They name this new recipe as REWARDMAP which shows performance gains on the test set of REASONMAP and gains of around 3.5% on other reasoning benchmarks as well.

**Strengths:**

1. The paper shows gains using REWARDMAP on both in-distribution and out of distribution test sets, suggesting that the curriculum learning + denser reward works well for multi-modal reasoning training
2. They construct an augmented dataset REASONMAP-PLUS with around 4k VQA pairs across different types like local counting, global counting, and true/false
3. The difficulty and dense rewards are assigned programatically without the need for any reward models or human labelling which makes the method cheap and efficient

**Weaknesses:**

1. Incremental novelty: both curriculum learning (https://arxiv.org/abs/2506.06632, https://arxiv.org/abs/2501.12599, https://arxiv.org/abs/2502.14768) and denser rewards to improve reasoning are approaches which have been tried in several existing works, while this paper shows a combination which works well, the approach as a whole feels incremental.
2. Reward shaping not generalizable: The detail reward (credit for route names, transfers) is tightly linked to transit-map structure; it’s unclear how the same scheme would extend to charts, floor plans, or natural images without such information. Likewise, the difficulty weights rely on a dataset-specific notion of map/question difficulty (e.g., transfer counts). It would be good to show a more generalizable approach to reward design.
3. Reward shaping not ablated well: The reward seems ad-hoc, there are no clear ablations for how the value of alpha was chosen, nor are there ablations on how W_diffculty is decided, or why it is multiplied to the other rewards. It would be good to provide some additional experimental results as to how this particular reward structure was chosen.
4. Limited OOD improvements: While the authors show performance on other benchmarks like MM-Star as well, the performance on them shows negligible gains. It might be more informative if there is a task wise breakdown of the performance which might help analyze the exact tasks where performance improves vs where it stagnates.

**Questions:**

Please refer to the weakness section for questions.

---

> ### Author Response · Authors · 2025-11-21
> **Response to Reviewer ekjh (Part-1)**
>
> We sincerely thank Reviewer `ekjh` for the helpful and insightful feedback. We appreciate the reviewer’s recognition of our method and dataset. Below, we respond to the suggestions and outline the improvements we have implemented, with all revisions highlighted in **blue**.
>
> ---
>
> > **W1**: *About the technical novelty of the method.*
>
> **A**: Thank you for the comments. We agree that curriculum learning and the use of denser rewards have been explored in several recent works on LLM reasoning (e.g., the references kindly pointed out by the reviewer). These studies provide valuable insights, and we have discussed them in more detail in the paper.
>
> Our contribution, however, lies in extending these ideas to the multimodal setting, where the challenges are fundamentally different. Fine-grained visual reasoning introduces additional sources of instability, including visual confusion, grounding errors, and sparse perceptual rewards, which are absent from purely textual tasks. To our best knowledge, our work is the first to integrate structured dense supervision with a multi-stage RL pipeline tailored specifically for VLMs, enabling them to acquire both perceptual and reasoning competencies in a coordinated manner. This design leads to substantial gains on ReasonMap and ReasonMap-Plus, as well as consistent improvements across diverse external benchmarks, suggesting that the framework addresses limitations unique to visual reasoning under RL.
>
> We appreciate the reviewer’s perspective and have incorporated further discussion to clarify how our approach builds upon and differs from prior work (see Section 2 of the updated manuscript).
>
> ---
>
> > **W2**: *About the generalization of reward shaping.*
>
> **A**: Thank you for raising this insightful concern. While our detail reward is indeed tailored to the structure of transit maps, the underlying idea—augmenting supervision with task-specific intermediate signals—is general and can be readily adapted to other structured visual domains such as diagrams or charts. The resulting improvements are not limited to map navigation: our framework yields substantial gains of 18.23%–30.04% on the primary tasks (ReasonMap and ReasonMap-Plus), as well as consistent improvements (average 3.47%) across six external benchmarks, indicating a broader enhancement in visual reasoning capability. More broadly, our results also highlight the value of structured visual data, such as transit maps, as an effective post-training supervision for strengthening multimodal reasoning skills. We appreciate the reviewer’s thoughtful feedback.
>
> ---
>
> > **W4**: *About the OOD improvements.*
>
> **A**: Thank you for the constructive suggestion. Although the gains on external benchmarks may appear modest, they are nonetheless meaningful. The six datasets in Table 2 serve as out-of-distribution (OOD) evaluations, and we observe consistent improvements across all of them using VLMEvalKit to ensure fair assessment. Given that modern MLLMs are already close to saturation on these benchmarks, even small gains are widely considered significant—an observation also highlighted in recent technical reports such as Qwen2.5-VL[`R1`] and InternVL3[`R2`].
>
> We further provide a task-wise breakdown of the performance on the six OOD benchmarks (where applicable): (1) SEED-Bench-2-Plus: We report the results on its map split; (2) SpatialEval: Accuracy on the mazenav task increases from 19.60% to 57.20%, while the spatialreal task improves from 70.37% to 72.59%; (3) HRBench: Accuracies improve from 85.25% to 88.00% and 52.25% to 54.00%, respectively. These correspond to single-image reasoning and cross-image alignment; (4) MMStar: We provide results on its key subtasks, including fine-grained perception (59.60% to 60.80%), instance reasoning (68.00% to 71.20%), and math (59.60% to 63.20%).
>
> Overall, these task-wise results reveal several consistent patterns. Improvements on fine-grained perception, instance reasoning, and real-world spatial understanding suggest that RewardMap strengthens the model’s fundamental visual grounding capabilities. Meanwhile, gains in more challenging settings—such as maze navigation, cross-image reasoning, and map-based questions—indicate enhanced robustness in structurally complex or multi-step spatial tasks, even when absolute accuracy remains modest. Together, these findings provide a more granular view of the OOD benefits and confirm that the improvements are not confined to a single task type but distributed across diverse categories of visual reasoning.
>
> We appreciate the reviewer’s helpful feedback, and the manuscript has been updated with the corresponding analyses (refer to Section 5.2).
>
> ---
>
> **References:**
>
> - [`R1`] Bai, Shuai, et al. "Qwen2.5-VL technical report." arXiv preprint arXiv:2502.13923 (2025).
> - [`R2`] Zhu, Jinguo, et al. "InternVL3: Exploring advanced training and test-time recipes for open-source multimodal models." arXiv preprint arXiv:2504.10479 (2025).

---

> > ### Author Response · Authors · 2025-11-21
> > **Response to Reviewer ekjh (Part-2)**
> >
> > > **W3**: *Suggestion on adding more detailed analysis.*
> >
> > **A**: Thank you for raising this important concern. The reward components and their hyperparameters are not key design to our method, and in the original experiments we adopted moderate default values: α = 0.5, γ_e/m/h = [1.0, 1.2, 1.5], and β_0/1 = [0.0, 0.5]. To address the reviewer’s request for clearer ablations, we conducted three additional sets of studies on α, γ_e/m/h, and β_0/1 as detailed below.
> >
> > 1. Based on the full RewardMap pipeline, we varied only α. The results are provided in the table. We observe a slight performance drop when α is very small, while performance becomes stable and comparable across models once α enters a moderately large range. This indicates that the detailed reward signal is indeed helpful (as α controls its relative contribution) and that α is not sensitive within a reasonable interval.
> > | Model | Training Data | ReasonMap Weighted Acc. (S./L.) | ReasonMap Weighted Map Score (S./L.) | ReasonMap-Plus Weighted Acc. | ReasonMap-Plus Weighted Acc. (Count / TorF) |
> > |-|-|-|-|-|-|
> > | Qwen2.5-VL-7B-Instruct | - | 13.28% / 7.12% | 4.01 / 5.74 | 44.21% | 37.39% / 51.32%  |
> > | + RewardMap [α=0.3]   | ReasonMap_train + ReasonMap_Plus_train | 30.73% / 31.16% | 6.01 / 10.81 | 72.08% | 68.41% / 75.91%  |
> > | + RewardMap [α=0.5]   | ReasonMap_train + ReasonMap_Plus_train | 31.51% / 31.77% | 6.21 / 11.22 | 74.25% | 72.18% / 76.42% |
> > | + RewardMap [α=0.7]   | ReasonMap_train + ReasonMap_Plus_train | 32.03% / 32.20% | 6.23 / 11.20 | 72.81% | 70.14% / 75.59% |
> >
> > 2. Using the ReasonMap training set, we further provide two complementary ablations: one varying only the relative magnitudes of γ_e/m/h, and another varying only the relative magnitudes of β_0/1. As shown in the tables, performance declines slightly when their relative weights become nearly uniform, but remains stable when the difficulty levels are distinctly weighted. These results support the effectiveness of the difficulty-aware components—since γ_e/m/h and β_0/1 control difficulty weighting for map difficulty and question difficulty respectively—and demonstrate that both hyperparameters are robust within a broad range.
> > | Model | Training Data | ReasonMap Weighted Acc. (S./L.) | ReasonMap Weighted Map Score (S./L.) | ReasonMap-Plus Weighted Acc. | ReasonMap-Plus Weighted Acc. (Count / TorF) |
> > |-|-|-|-|-|-|
> > | Qwen2.5-VL-7B-Instruct | - | 13.28% / 7.12% | 4.01 / 5.74 | 44.21% | 37.39% / 51.32%  |
> > | + Reward Design [γ_e/m/h = (1.0, 1.1, 1.2)] | ReasonMap_train | 26.91% / 28.47% | 5.64 / 10.24 | 45.34% | 37.97% / 53.02%  |
> > | + Reward Design [γ_e/m/h = (1.0, 1.2, 1.5)] | ReasonMap_train | 29.08% / 29.95% | 5.88 / 10.53 | 45.16% | 37.79% / 52.84%  |
> > | + Reward Design [γ_e/m/h = (1.0, 1.5, 2.0)] | ReasonMap_train | 29.86% / 28.99% | 5.93 / 10.31 | 45.16% | 37.31% / 53.35%  |
> >
> > | Model | Training Data | ReasonMap Weighted Acc. (S./L.) | ReasonMap Weighted Map Score (S./L.) | ReasonMap-Plus Weighted Acc. | ReasonMap-Plus Weighted Acc. (Count / TorF) |
> > |-|-|-|-|-|-|
> > | Qwen2.5-VL-7B-Instruct | - | 13.28% / 7.12% | 4.01 / 5.74 | 44.21% | 37.39% / 51.32%  |
> > | + Reward Design [β_0/1 = (0.0, 0.2)] | ReasonMap_train | 28.30% / 29.34% | 5.68 / 10.12 | 44.60% | 37.39% / 52.10%  |
> > | + Reward Design [β_0/1 = (0.0, 0.5)] | ReasonMap_train | 29.08% / 29.95% | 5.88 / 10.53 | 45.16% | 37.79% / 52.84%  |
> > | + Reward Design [β_0/1 = (0.0, 0.8)] | ReasonMap_train | 29.60% / 29.34% | 5.89 / 10.23 | 45.39% | 38.68% / 52.38%  |
> >
> > We thank the reviewer again for the constructive suggestion, and we have added these ablations and analyses to the manuscript (see Section 5.4).
> >
> > ---
> >
> > Last but not least, we would like to sincerely thank Reviewer `ekjh` again for the valuable time and constructive feedback provided during this review.
> >
> > **Thank you for helping improve our work so far! We are actively available during the discussion period. Let us know should you have any further questions.**

---

> > > ### Comment · Reviewer_ekjh · 2025-11-24
> > >
> > > Thanks, I have updated my score based on the experiments and discussion.

---

> > > > ### Author Response · Authors · 2025-11-24
> > > > **Sincere Thanks to Reviewer ekjh**
> > > >
> > > > Dear Reviewer `ekjh`,
> > > >
> > > > We sincerely thank you for revisiting our work and raising the score. We appreciate the reviewer’s time and effort in carefully examining our new experiments and extended analyses.
> > > >
> > > > Thank you again for the constructive feedback throughout the process. We are still active during the discussion process and would be happy to provide any additional clarification should you have further thoughts.
> > > >
> > > > Best regards,
> > > >
> > > > The Authors of Submission 397

---

### Official Review · Reviewer_owjV · 2025-10-28

**Soundness:** 3
**Presentation:** 3
**Contribution:** 2
**Rating:** 4
**Confidence:** 3

**Summary:**

This paper studys the problem of sparse reward in standard reinforcement learning for spatial reasoning in structured and information-rich setting.  To address the problem, the authors construct REASONMAP-PLUS, a dataset with dense reward signals and propose a multi-stage RL framework for improving the corresponding capabilities in current MLLM. Experiment results demonstrate its effectiveness across various reasoning tasks and general tasks.

**Strengths:**

1. The research topic of this work is interesting, as sparse reward is indeed a significant challenge in current vision tasks, particularly in structured visual input tasks as highlighted in the paper.
2. This work introduces the REASONMAP-PLUS dataset, which could be beneficial for future related research.
3. The experimental results seem promising, demonstrating the effectiveness of the proposed method on transit map planning tasks and showing its transferability to other tasks.

**Weaknesses:**

1. The method parts of the paper presents certain ambiguities. For instance, how does the Difficulty-Aware Weighting mechanism operate? Are weights assigned to each sample within the groups of GRPO, or to each sample across the entire training dataset? Furthermore, what does Multi-stage RL training entail—does it refer solely to the sequencing of data based on difficulty, or does it also encompass different training phases?
2. The novelty of the approach appears limited, as the core methodology remains largely aligned with that of GRPO. While the design of the detailed reward function is interesting, it only considers specific components of the response (e.g., destination stops, route names), thereby constraining its overall flexibility.

**Questions:**

1. In the acquisition of the detailed reward, how is the correctness of each sub-part determined? Is an external LLM employed to extract the relevant content, or is the response constrained to a fixed format?
2. Please refer to the weaknesses highlighted above.

---

> ### Author Response · Authors · 2025-11-21
> **Response to Reviewer owjV**
>
> We sincerely thank Reviewer `owjV` for the thoughtful and insightful feedback. We appreciate the reviewer’s recognition of our topic, dataset, and performance. Below, we address the suggestions and summarize the improvements we have incorporated, with all revisions marked in **blue**.
>
> ---
>
> > **W1**: *Clarification on the difficulty-aware weighting and multi-stage RL mechanism.*
>
> **A**: Thank you for this constructive comment. Firstly, regarding the difficulty-aware weighting strategy, weights are assigned to individual training samples. However, because the advantage is computed based on the relative rewards of samples within a group, the weighting indirectly influences optimization by increasing the contribution of harder samples during training. Secondly, our multi-stage design is not simply a global sorting of all training data by map or question difficulty. Instead, we first partition the data into several global stages, each corresponding to different task types (e.g., from binary judgment to counting to planning, and from visual understanding to visual reasoning). Within each stage, samples are shuffled rather than arranged in strict difficulty order. In addition, each stage adopts a different reward composition: for instance, binary judgment tasks use only accuracy and format rewards, whereas planning tasks incorporate the detail reward terms described in the paper.
>
> ---
>
> > **W2**: *About the novelty and core methodology.*
>
> **A**: Thank you for raising this important concern. Our work aims to make RL effective for fine-grained visual reasoning, a setting where existing RL pipelines often struggle due to sparse rewards and unstable optimization. To tackle these challenges, we introduce ReasonMap-Plus to provide dense supervision for RL training, and we design a multi-stage RL framework that aligns task progression with the acquisition of perceptual and reasoning skills. While our detail reward is tailored to the structure of the primary task, the idea of enriching supervision with task-specific intermediate signals is general and can be readily adapted to other structured visual domains (e.g., diagrams, charts). These components collectively yield substantial improvements of 18.23%–30.04% on our main tasks (ReasonMap & ReasonMap-Plus), as well as consistent gains (average 3.47%) across six external benchmarks, indicating enhanced general visual reasoning capability. We hope that our dataset and training pipeline can serve as a useful reference for future work on applying RL to structured visual tasks.
>
> ---
>
> > **Q1**: *Clarification about reward computation details.*
>
> **A**: Thank you for raising this important question. We do not rely on any external LLMs for answer extraction; instead, all outputs follow a fixed format. For ReasonMap, we adhere to its original evaluation pipeline, which is based on parsing and string matching with normalization to remove stylistic variations (e.g., ‘Route 1’, ‘Line 1’, and ‘1’ are treated as equivalent), ensuring precise reward computation. For ReasonMap-Plus, correctness is also determined by string matching, since answers are single numbers or letters in a fixed format (e.g., ‘\boxed{A}’, ‘\boxed{12}’). This evaluation procedure is consistent with widely-used reasoning benchmarks such as GSM8K and MATH-500. We thank the reviewer again for pointing out this notable question (refer to Section 4.2 for more details).
>
> ---
>
> Last but not least, we would like to sincerely thank Reviewer `owjV` again for the valuable time and constructive feedback provided during this review.
>
> **Thank you for helping improve our work so far! We are actively available during the discussion period. Let us know should you have any further questions.**

---

> > ### Comment · Reviewer_owjV · 2025-11-24
> >
> > The rebuttal has partially addressed my concerns. However, I would appreciate it if the authors could provide a concrete dataset sample that includes additional process-level annotations, as this would facilitate a better understanding of the supplementary contributions of this work in terms of dataset annotation. I would be willing to raise my score if such an example is provided.

---

> > > ### Author Response · Authors · 2025-11-25
> > > **Further Response to Reviewer owjV**
> > >
> > > We sincerely thank the reviewer for reading our rebuttal and revisiting our work. We address the suggestion and make the corresponding revisions. We address the suggestion below and summarize the corresponding revisions, which have been incorporated into the manuscript and marked in **blue**.
> > >
> > > ---
> > >
> > > **Q**: *Suggestion on providing a concrete dataset sample including additional process-level annotations.*
> > >
> > > **A**: We thank the reviewer for the follow-up request. We also provide fine-grained annotations beyond the final answer. The annotation process is as follows. We first annotate the route–stop information on each map, including all routes, their stops, and key interchange or branching stations (as shown in the Meta Data below).
> > >
> > > ```
> > > The Meta Data (Singapore)
> > >
> > > {
> > > 	"Circle Line": ["HarbourFront (Transfer)", "Telok Blangah", "Labrador Park", "Pasir Panjang", ...
> > > 	],
> > >
> > > 	"Downtown Line": [...],
> > >
> > > 	...
> > > }
> > > ```
> > >
> > > Based on these pre-annotations, we then generate process-level annotations using task-specific Python scripts. This annotation pipeline makes the construction easy to extend. For example, in the ‘Local Counting 1’ category of the Counting tasks in ReasonMap-Plus, where the question asks for the number of intermediate stops between two given stations, we first locate the two stations in the Meta Data. We then identify all intermediate stops, compute their count as the final answer, and record the intermediate stops themselves as the process-level annotation. Other task types follow a similar annotation procedure. We present two examples as follows ("fine-grained answer" denotes the process-level annotations).
> > >
> > > ```
> > > Example 1 (Local Counting 1)
> > >
> > > "country": "singapore",
> > > "city": "singapore",
> > > "station_1": "Clarke Quay",
> > > "station_2": "Farrer Park",
> > > "figure": "./maps/singapore/singapore.png",
> > > "question": "According to the map, how many intermediate stops are there between Clarke Quay and Farrer Park (except for these two stops)?",
> > > "answer": "2",
> > > "fine-grained answer": ["Dhoby Ghaut", "Little India"],
> > > "type": "counting_1",
> > > "difficulty_city": "hard",
> > > "city_line_count": "6",
> > > "city_transfer_count": "53",
> > > "json": "./stations/singapore/singapore.json", (contains the Meta Data)
> > > ```
> > >
> > > ```
> > > Example 2 (True or False 1)
> > >
> > > "country": "singapore",
> > > "city": "singapore",
> > > "station_1": "Somerset",
> > > "station_2": "Bras Basah",
> > > "figure": "./maps/singapore/singapore.png",
> > > "question": "According to the map, is it true that Somerset is on the same line as Bras Basah?",
> > > "answer": "No",
> > > "fine-grained answer": {"Somerset": "North South Line", "Bras Basah": "Circle Line"},
> > > "type": "torf_1",
> > > "difficulty_city": "hard",
> > > "city_line_count": "6",
> > > "city_transfer_count": "53",
> > > "json": "./stations/singapore/singapore.json", (contains the Meta Data)
> > > ```
> > >
> > > We thank the reviewer again for this valuable suggestion. We have revised the manuscript accordingly, as reflected in Appendix A.3.

---

> > > > ### Comment · Reviewer_owjV · 2025-11-25
> > > >
> > > > After reviewing the rebuttal part, I think the author has addressed most of my concerns and I acknowledge the solidity of this study. I decide to raise my score.

---

> > > > > ### Author Response · Authors · 2025-11-25
> > > > > **Sincere Thanks to Reviewer owjV**
> > > > >
> > > > > Dear Reviewer `owjV`,
> > > > >
> > > > > Thank you very much for revisiting our work and for deciding to raise the score. We truly appreciate your **positive recognition** of the overall solidity of our study, which is very encouraging to us.
> > > > >
> > > > > We are also grateful for the time and care you devoted to reading our rebuttal and for your constructive feedback throughout the process.
> > > > >
> > > > > Best regards,
> > > > >
> > > > > The Authors of Submission 397

---

### Official Review · Reviewer_GZ1U · 2025-10-30

**Soundness:** 4
**Presentation:** 3
**Contribution:** 3
**Rating:** 8
**Confidence:** 3

**Summary:**

This paper proposes RewardMap, a multi-stage RL framework that mitigates sparse-reward problems in fine-grained visual reasoning for multimodal LLMs. Building on ReasonMap, the authors introduce ReasonMap-Plus, a dense and difficulty-graded dataset for visual question answering on transit maps. RewardMap integrates (1) a difficulty-aware reward function combining format, correctness, and detail rewards, and (2) a multi-stage GRPO training curriculum progressing from perception to reasoning tasks. Experiments on RewardMap, ReasonMap-Plus, and six external benchmarks show consistent gains and reduced visual confusion, demonstrating improved visual understanding and reasoning performance.

**Strengths:**

1. New dataset and training paradigm: the paper introduces ReasonMap-Plus, which is a novel dataset that helps conducting RL using dense reward signals.
2. Models being trained on the dataset seem to perform well on the proposed benchmarks

**Weaknesses:**

1. Lack of baseline VLMs on ReasonMap and ReasonMap-Plus. What would more recent and powerful VLMs like GPT-4o and GPT-5 perform on these benchmarks?
2. Would be better to train other VLMs other than Qwen2.5-VL, such as InternVL models to see if the training and data could actually be generalized.

**Questions:**

N/A

---

> ### Author Response · Authors · 2025-11-21
> **Response to Reviewer GZ1U**
>
> We sincerely thank Reviewer `GZ1U` for the thoughtful and insightful suggestion. We highly appreciate the reviewer’s recognition of our dataset, method, and performance. Below, we address the comments in detail and summarize the corresponding revisions, which have been updated in the manuscript and highlighted in **blue**.
>
> ---
>
> > **W1**: *Suggestion for evaluation on powerful closed-source models.*
>
> **A**: Thank you for this insightful comment. Our goal is only to provide reference baselines by including results from representative open- and closed-source models. In response to the reviewer’s suggestion, we further evaluated GPT-4o and GPT-5 on ReasonMap and ReasonMap-Plus, following the evaluation frameworks of each benchmark. The results are shown below. We appreciate the reviewer’s constructive suggestion and have updated the manuscript accordingly, including the new results in Table 1.
>
> | Model | ReasonMap Weighted Acc. (S./L.) | ReasonMap Weighted Map Score (S./L.) | ReasonMap-Plus Weighted Acc. | ReasonMap-Plus Weighted Acc. (Count / TorF) |
> |-|-|-|-|-|
> | GPT-4o | 41.15% / 42.80% | 6.84 / 13.57 | 64.42% | 59.28% / 69.77%  |
> | GPT-5  | 59.98% / 62.50% | 9.48 / 19.75 | 88.95% | 86.40% / 91.60%  |
>
> > **W2**: *Suggestion for testing generalization across model architectures.*
>
> **A**: Thank you for this constructive suggestion. As is common in prior work, we initially focused on a single model family due to limited computational resources. To address the reviewer‘s suggestion, we further conducted experiments on Kimi VL (Kimi-VL-A3B-Instruct) using the RewardMap training pipeline, and the results are shown below. RewardMap achieves substantial performance improvements for Kimi VL, further demonstrating the cross-model generalization of our method and dataset. We again thank the reviewer for the valuable feedback, and we have updated the manuscript accordingly (see Section 5.4).
>
> | Model | Training Data |ReasonMap Weighted Acc. (S./L.) | ReasonMap Weighted Map Score (S./L.) | ReasonMap-Plus Weighted Acc. | ReasonMap-Plus Weighted Acc. (Count / TorF) |
> |-|-|-|-|-|-|
> | Kimi-VL-A3B-Instruct | - |12.76% / 12.33% | 3.30 / 5.37 | 32.55% | 14.75% / 51.08%  |
> | + RewardMap  | ReasonMap_train | 18.58% / 17.36% | 4.70 / 7.69 | 35.92% | 15.20% / 57.50%  |
>
> ---
>
> Last but not least, we would like to sincerely thank Reviewer `GZ1U` again for the valuable time and constructive feedback provided during this review.
>
> **Thank you for helping improve our work so far! We are actively available during the discussion period. Let us know should you have any further questions.**

---

### Official Review · Reviewer_3FQv · 2025-11-01

**Soundness:** 2
**Presentation:** 3
**Contribution:** 3
**Rating:** 6
**Confidence:** 4

**Summary:**

This paper proposes REWARDMAP, a multi-stage reinforcement learning (RL) framework for improving fine-grained visual reasoning in multimodal large language models (MLLMs). It builds on the REASONMAP benchmark and introduces REASONMAP-PLUS, a companion dataset providing dense supervision via VQA-style tasks organized by difficulty. The approach combines a difficulty-aware reward design with curriculum-based RL under GRPO to mitigate sparse reward issues. Experiments show consistent but small gains (around 3.5%) on REASONMAP, REASONMAP-PLUS, and several external benchmarks. Ablations suggest that reward shaping and staged training both contribute to improvements.

**Strengths:**

- Addresses a critical problem of sparse rewards in RL for multimodal reasoning with a practical and elegant solution.
- Well-engineered multi-stage RL approach combining dense and sparse supervision effectively.
- Comprehensive evaluation with ablations that isolate reward design and curriculum effects.
- Clear writing, reproducible setup, and solid empirical results across multiple benchmarks.

**Weaknesses:**

- Conceptual novelty is limited. The method is primarily a well-engineered combination of known ideas: reward shaping, difficulty weighting, and curriculum learning under GRPO. There is no new RL algorithm or theoretical contribution.
- Reported performance gains are numerically small relative to the additional dataset, compute, and engineering effort. It is unclear whether such modest improvements are statistically significant.
- The reward weighting scheme (α, γ_e/m/h, β_0/1) is ad-hoc and lacks sensitivity analysis. Without this, claims about “difficulty awareness” remain anecdotal.
- Evaluation is restricted to Qwen2.5-VL, raising concerns about generalization and overfitting to a single model architecture.
- REASONMAP-PLUS is narrowly scoped to transit maps; claims of general fine-grained reasoning ability are not convincingly supported.
- The paper lacks deeper insight or analysis into why multi-stage RL works, there is no visualization or error decomposition that would make the mechanism interpretable.
- The average improvement on external benchmarks (+3.47%) is within noise for modern MLLMs; effect size is weak.

**Questions:**

1. How sensitive are results to the hyperparameters \( \alpha \) and difficulty weights (\( \gamma_e, \gamma_m, \gamma_h \))?
2. Could the proposed curriculum RL pipeline be applied to other structured domains (e.g., diagrams, charts)?
3. How do you ensure no data leakage between REASONMAP-PLUS easy/hard tasks and REASONMAP test sets?
4. Why were RLHF or DPO-based baselines not compared for reward densification?
5. Can you provide error-type or reasoning-chain analyses to explain where REWARDMAP helps most?

**Details Of Ethics Concerns:**

No significant ethical issues were found. The dataset is built from public transit maps and poses minimal privacy or safety risks. Potential copyright concerns should be clarified, but usage for research likely falls under fair use.

---

> ### Author Response · Authors · 2025-11-21
> **Response to Reviewer 3FQv (Part-1)**
>
> We sincerely thank Reviewer `3FQv` for the thoughtful and constructive feedback. We are pleased that the reviewer found our method effective and appreciated the comprehensive evaluation. Below, we address the reviewer’s suggestions and outline the improvements we have incorporated moving forward, and we have revised the manuscript accordingly with updates marked in **blue**.
>
> ---
>
> > **W1:** *About the conceptual novelty.*
>
> **A**: Thank you for raising this important concern. Our work focuses on making RL truly effective for fine-grained visual reasoning, a setting where existing RL pipelines often struggle due to **sparse rewards and unstable optimization**. To address these challenges, we introduce several key components. First, we develop ReasonMap-Plus to provide dense and fine-grained supervision for RL training. Second, we design a multi-stage RL framework that aligns task progression with the gradual acquisition of perceptual and reasoning skills. Third, these components together yield substantial improvements of 18.23%–30.04% on our primary tasks (ReasonMap and ReasonMap-Plus), along with consistent gains (average 3.47%) across six external datasets, demonstrating enhanced general visual reasoning ability. We hope that our dataset and training pipeline can serve as a useful reference for future research on applying RL to structured visual tasks.
>
> ---
>
> > **W2 & W7**: *About the improvements on additional dataset.*
>
> **A**: Thank you for the helpful concern. Firstly, the performance gains on our primary evaluation targets (ReasonMap & ReasonMap-Plus) are substantial, ranging from 18.23% to 30.04%. Secondly, the six benchmarks in Table 2 serve as **out-of-distribution (OOD) evaluations**, where we observe consistent improvements across all datasets using VLMEvalKit for fair evaluation. Given that modern MLLMs are already close to saturation on these benchmarks, even modest gains are widely regarded as meaningful, as reflected in recent technical reports such as Qwen2.5-VL[`R1`] and InternVL3[`R2`]. In addition to these moderate but reliable gains, we also observe large improvements on challenging benchmarks such as SpatialEval (+13.51%).
>
> ---
>
> > **W4**: *Suggestion of extending to other model architectures.*
>
> **A**: Thank you for raising this insightful concern. As in prior work, most RL-based studies typically focus on a single model family, and our initial experiments followed this practice due to computational constraints. In response to the reviewer’s concern, we additionally trained Kimi VL (Kimi-VL-A3B-Instruct) with the RewardMap pipeline. The results, presented below, show substantial gains, demonstrating that our method generalizes beyond the Qwen2.5-VL series. We thank the reviewer again for the helpful suggestion and have updated the manuscript accordingly (see Section 5.4).
>
> | Model | Training Data |ReasonMap Weighted Acc. (S./L.) | ReasonMap Weighted Map Score (S./L.) | ReasonMap-Plus Weighted Acc. | ReasonMap-Plus Weighted Acc. (Count / TorF) |
> |-|-|-|-|-|-|
> | Kimi-VL-A3B-Instruct | - |12.76% / 12.33% | 3.30 / 5.37 | 32.55% | 14.75% / 51.08%  |
> | + RewardMap  | ReasonMap_train | 18.58% / 17.36% | 4.70 / 7.69 | 35.92% | 15.20% / 57.50%  |
>
> ---
>
> > **W5**: *Clarification about the claims of general fine-grained reasoning.*
>
> **A**: Thank you for pointing out this helpful question. Our work claims that ReasonMap-Plus serves as an auxiliary dataset focused on visual understanding, as noted in Section 3.1, and it is used only in the early stages of our training pipeline. Our claims of general fine-grained visual reasoning are supported by consistent gains not only on ReasonMap, which requires detailed visual understanding and spatial reasoning for route planning, but also on popular fine-grained reasoning benchmarks such as V* and HRBench, showing that the improvements extend beyond a single dataset.

---

> > ### Author Response · Authors · 2025-11-21
> > **Response to Reviewer 3FQv (Part-2)**
> >
> > > **W3 & Q1**: *Suggestion of adding more ablation study.*
> >
> > **A**: Thank you for the valuable feedback. The hyperparameters in this section are not critical to our method. In the original experiments, we adopted moderate default values: α = 0.5, γ_e/m/h = [1.0, 1.2, 1.5], and β_0/1 = [0.0, 0.5]. To address the reviewer’s concern, we additionally conducted three sets of ablation studies on α, γ_e/m/h, and β_0/1 as follows.
> >
> > 1. Based on the full RewardMap pipeline, we varied only α. The results are reported as follows. We observe a slight performance drop when α is small, while the performance becomes stable once α enters a higher range. These results suggest that the detailed reward is indeed beneficial (as α determines its contribution to the total reward), and they also indicate that α is not sensitive within a reasonable interval.
> >
> > | Model | Training Data | ReasonMap Weighted Acc. (S./L.) | ReasonMap Weighted Map Score (S./L.) | ReasonMap-Plus Weighted Acc. | ReasonMap-Plus Weighted Acc. (Count / TorF) |
> > |-|-|-|-|-|-|
> > | Qwen2.5-VL-7B-Instruct | - | 13.28% / 7.12% | 4.01 / 5.74 | 44.21% | 37.39% / 51.32%  |
> > | + RewardMap [α=0.3]   | ReasonMap_train + ReasonMap_Plus_train | 30.73% / 31.16% | 6.01 / 10.81 | 72.08% | 68.41% / 75.91% |
> > | + RewardMap [α=0.5]   | ReasonMap_train + ReasonMap_Plus_train | 31.51% / 31.77% | 6.21 / 11.22 | 74.25% | 72.18% / 76.42% |
> > | + RewardMap [α=0.7]   | ReasonMap_train + ReasonMap_Plus_train | 32.03% / 32.20% | 6.23 / 11.20 | 72.81% | 70.14% / 75.59% |
> >
> > 2. Using the ReasonMap training set, we conducted two additional experiments: one varying only the relative magnitudes of γ_e/m/h, and another varying only the relative magnitudes of β_0/1. The results are shown in Table X. For both γ_e/m/h and β_0/1, we observe that performance drops slightly when their relative weights become nearly uniform, while performance remains comparable when the differences in difficulty levels are clearly expressed. This supports the effectiveness of our difficulty-aware design—since γ_e/m/h and β_0/1 respectively control the difficulty weighting for map difficulty and question difficulty—and also shows that both hyperparameters are insensitive within a reasonable range.
> >
> > | Model | Training Data | ReasonMap Weighted Acc. (S./L.) | ReasonMap Weighted Map Score (S./L.) | ReasonMap-Plus Weighted Acc. | ReasonMap-Plus Weighted Acc. (Count / TorF) |
> > |-|-|-|-|-|-|
> > | Qwen2.5-VL-7B-Instruct | - | 13.28% / 7.12% | 4.01 / 5.74 | 44.21% | 37.39% / 51.32%  |
> > | + Reward Design [γ_e/m/h = (1.0, 1.1, 1.2)] | ReasonMap_train | 26.91% / 28.47% | 5.64 / 10.24 | 45.34% | 37.97% / 53.02%  |
> > | + Reward Design [γ_e/m/h = (1.0, 1.2, 1.5)] | ReasonMap_train | 29.08% / 29.95% | 5.88 / 10.53 | 45.16% | 37.79% / 52.84%  |
> > | + Reward Design [γ_e/m/h = (1.0, 1.5, 2.0)] | ReasonMap_train | 29.86% / 28.99% | 5.93 / 10.31 | 45.16% | 37.31% / 53.35%  |
> >
> > | Model | Training Data | ReasonMap Weighted Acc. (S./L.) | ReasonMap Weighted Map Score (S./L.) | ReasonMap-Plus Weighted Acc. | ReasonMap-Plus Weighted Acc. (Count / TorF) |
> > |-|-|-|-|-|-|
> > | Qwen2.5-VL-7B-Instruct | - | 13.28% / 7.12% | 4.01 / 5.74 | 44.21% | 37.39% / 51.32%  |
> > | + Reward Design [β_0/1 = (0.0, 0.2)] | ReasonMap_train | 28.30% / 29.34% | 5.68 / 10.12 | 44.60% | 37.39% / 52.10%  |
> > | + Reward Design [β_0/1 = (0.0, 0.5)] | ReasonMap_train | 29.08% / 29.95% | 5.88 / 10.53 | 45.16% | 37.79% / 52.84%  |
> > | + Reward Design [β_0/1 = (0.0, 0.8)] | ReasonMap_train | 29.60% / 29.34% | 5.89 / 10.23 | 45.39% | 38.68% / 52.38%  |
> >
> > We thank the reviewer again for the constructive suggestion, and the manuscript has been updated accordingly (see Section 5.4).
> >
> > ---
> >
> > > **Q2**: *Question about extending the RewardMap pipeline to other structured domains.*
> >
> > **A**: Thank you for the thoughtful suggestion. Our RL pipeline is not limited to transit maps and can be naturally applied to other structured visual domains. Firstly, the core ideas behind our method (e.g., multi-stage RL scheme and task-specific detail rewards) are both general and extendable to other structured domains like charts or diagrams. Secondly, our approach also improves performance on other structured benchmarks (e.g., ChartQA in Table 1 and the additional benchmarks below) under the out-of-distribution (OOD) setting, indicating that the learning signals from maps can be transferred to structurally similar domains, further suggesting an underlying correspondence across structured visual tasks. We highly appreciate the reviewer’s insightful comment, and we have updated the manuscript adding the additional results and a future work section to explore this direction explicitly (refer to Appendix E).
> >
> > | Model | ChartQA | Charxiv (Descriptive Questions)[`R3`] | InfoVQA[`R4`] |
> > |-|-|-|-|
> > | Qwen2.5-VL-7B-Instruct | 86.12% | 66.41% | 82.22% |
> > | + RewardMap            | 87.24% | 68.90% | 82.71% |

---

> ### Author Response · Authors · 2025-11-21
> **Response to Reviewer 3FQv (Part-3)**
>
> > **W6 & Q5**: *Suggestion on adding more detailed analysis.*
>
> **A**: Thank you for the constructive feedback. From the training dynamics, our analysis in Figure 4 shows clear differences in the reward signals with and without our method, suggesting that RewardMap helps alleviate the sparse-reward issue observed in baseline training. In addition, our case studies indicate that RewardMap notably reduces common failure modes in the original dataset—especially visual confusion and hallucination—as illustrated in Figure 3 and further discussed in Section 5.3. We also include additional comparison cases, which further show that RewardMap enhances visual grounding and lowers the risk of visual confusion or hallucination. We thank the reviewer for the valuable suggestion, and we have incorporated the additional examples and analyses into the manuscript (see Section 5.3 and Appendix D.1).
>
> Case 1 (The answer from Qwen2.5-VL-7B-Instruct shows a visual confusion error at *\<wrong1\>* and a hallucination at *\<wrong2\>*, incorrectly identifying three intermediate stops but selecting option A [6]):
> ```
> Q: How many intermediate stops are there between A Station and B Station? (<budapest.png> + Format Requirement + {A: 6, B: 8, C: 1, D: 9})
>
> A (Qwen2.5_VL_7B_Instruct, Wrong): To determine the number of intermediate stops between A Station and B Station, we need to look at the yellow line on the map, which represents Metro Line 1.\n\n1. Identify the stations on the yellow line between A and B.\n2. Count the number of stations, excluding A and B.\n\nFrom the map:\n- A\n- Széchenyi Thermal Bath\n- Bajza Street\n- B\n\nThere are 3 intermediate stops between A and B. <wrong1> \n\nTherefore, the correct answer is:\n\\boxed{A} <wrong2>
>
> A (Qwen2.5_vL_7B_Instruct_RewardMap, Correct): To solve this problem, we need to count the intermediate stops between A Station and B Station, excluding these two stops.\n\nStarting from A\n- The next stop is Széchenyi Thermal Bath.\n\nSo, the only intermediate stop between A and B is Széchenyi Thermal Bath.\n\nTherefore, the correct answer is:\n\\boxed{C}
> ```
>
> ---
>
> > **Q3**: *Question about how to ensure no data leakage.*
>
> **A**: Thank you for raising this important question. In ReasonMap, the train–test split is defined at the map level to ensure a **cross-city setting** and avoid data leakage (i.e., 11 maps for the test set and the remaining 19 maps for training). We follow the same split to keep leakage-free during the RewardMap training process. Thank the reviewer again for giving this comment (refer to Appendix A.1 for detailed statements).
>
> ---
>
> > **Q4**: *Suggestion on adding more baselines.*
>
> **A**: We appreciate this creative suggestion. We adopt GRPO as our primary RL algorithm because it has been shown to be particularly effective for reasoning tasks, offering more stable optimization compared to traditional policy-gradient methods. In response to the reviewer’s comment, we additionally conducted experiments with REINFORCE++[`R5`] and ReMax[`R6`] (i.e., these two methods are regarded as popular methods in RL training) to provide further baselines. Specifically, we trained all methods on the ReasonMap training set, and the results are presented below. We observe that REINFORCE++ and ReMax achieve slightly better performance than GRPO, yet both remain far below the effectiveness of our method. We thank the reviewer again for the helpful feedback, and we have updated the manuscript accordingly (see Table 1).
>
> | Model | Training Data |ReasonMap Weighted Acc. (S./L.) | ReasonMap Weighted Map Score (S./L.) | ReasonMap-Plus Weighted Acc. | ReasonMap-Plus Weighted Acc. (Count / TorF) |
> |-|-|-|-|-|-|
> | Qwen2.5-VL-7B-Instruct | - | 13.28% / 7.12% | 4.01 / 5.74 | 44.21% | 37.39% / 51.32%  |
> | + GRPO RL  | ReasonMap_train | 26.22% / 26.04% | 5.52 / 9.52 | 44.64% | 37.57% / 52.01%  |
> | + REINFORCE++ RL   | ReasonMap_train | 27.17% / 27.60% | 5.68 / 10.12 | 44.64% | 36.82% / 52.79% |
> | + ReMax RL   | ReasonMap_train | 26.22% / 27.26% | 5.57 / 9.99 | 45.39% | 38.37% / 52.70% |
> | + RewardMap  | ReasonMap_train + ReasonMap_Plus_train | 31.51% / 31.77% | 6.21 / 11.22 | 74.25% | 72.18% / 76.42% |

---

> > ### Author Response · Authors · 2025-11-21
> > **Response to Reviewer 3FQv (References)**
> >
> > **References:**
> >
> > - [`R1`] Bai, Shuai, et al. "Qwen2. 5-vl technical report." arXiv preprint arXiv:2502.13923 (2025).
> > - [`R2`] Zhu, Jinguo, et al. "Internvl3: Exploring advanced training and test-time recipes for open-source multimodal models." arXiv preprint arXiv:2504.10479 (2025).
> > - [`R3`] Wang, Zirui, et al. "Charxiv: Charting gaps in realistic chart understanding in multimodal llms." Advances in Neural Information Processing Systems 37 (2024): 113569-113697.
> > - [`R4`] Mathew, Minesh, et al. "Infographicvqa." Proceedings of the IEEE/CVF Winter Conference on Applications of Computer Vision. 2022.
> > - [`R5`] Hu, Jian, et al. "Reinforce++: An efficient rlhf algorithm with robustness to both prompt and reward models." arXiv preprint arXiv:2501.03262 (2025).
> > - [`R6`] Li, Ziniu, et al. "Remax: A simple, effective, and efficient reinforcement learning method for aligning large language models." arXiv preprint arXiv:2310.10505 (2023).
> >
> > ---
> >
> > Last but not least, we would like to sincerely thank Reviewer `3FQv` again for the valuable time and constructive feedback provided during this review.
> >
> > **Thank you for helping improve our work so far! We are actively available during the discussion period. Let us know should you have any further questions.**

---

### Author Response · Authors · 2025-12-01
**Summary of Rebuttal Process for the Area Chairs**

**Dear Area Chairs,**

Thank you very much for your time and careful reading of our manuscript and rebuttal. To support the newly assigned Area Chairs and ease their workload, we summarise key review points and discussions below.

---

**Rebuttal Process Overview** (Final score: *8664* by 25 Nov, 22:43):

| Reviewer / Time | 22 Nov, 01:00 | 24 Nov, 13:34 | 24 Nov, 22:06 | 25 Nov, 15:10 | 25 Nov, 22:43 | 27 Nov, 22:09 (Bug!) |
|-|-|-|-|-|-|-|
| `3FQv` (score 6) | 1st rebuttal | - | - | - | - | - |
| `GZ1U` (score 8) | 1st rebuttal | - | - | - | - | - |
| `owjV` (score 4 -> 6) | 1st rebuttal | - | acknowledge rebuttal & further ask | 2nd rebuttal | acknowledge solidity & raise to 6 | - |
| `ekjh` (score 2 -> 4) | 1st rebuttal | acknowledge rebuttal & raise to 4 | - | - | - | - |

---

We are encouraged by multiple points of consensus across reviewers:

- Reviewer `3FQv` (*score: 6*) acknowledged the contribution of mitigating sparse rewards in fine-grained visual reasoning and recognized the effectiveness of our multi-stage RL pipeline.
- Reviewer `GZ1U` (*score: 8*) highlighted the value of ReasonMap-Plus and noted the strong performance of RewardMap on structured spatial tasks.
- Reviewer `owjV` (*score raised from 4 -> 6 on 25 Nov, 22:43*) emphasized the importance of addressing sparse rewards in structured visual inputs, found ReasonMap-Plus beneficial for future research, and regarded our results as promising.
- Reviewer `ekjh` (*score raised from 2 -> 4 on 24 Nov, 13:34*) appreciated the dense VQA supervision, curriculum progression, and improvements in both in-domain and OOD settings.

---

We are also grateful that reviewers reconsidered their assessments after rebuttal and discussion:

- Reviewer `owjV` raised the score from **4 to 6** and 'acknowledged the solidity of this study' after our initial response resolved the main concerns, followed by a further increase after we provided process-level annotations of a concrete dataset sample.
- Reviewer `ekjh` raised the score from **2 to 4** after we added further ablations and deeper discussion.

---

In response to reviewer feedback, we made the following improvements and revisions:

- In response to  `3FQv` and `ekjh`, we added additional ablations to demonstrate stability. (integrated into Section 5.4)
- In response to  `3FQv` and `GZ1U`, we extended experiments to an additional model family (Kimi-VL), confirming that RewardMap generalizes beyond Qwen2.5-VL. (updated in Section 5.4)
- In response to `3FQv`, we incorporated more baselines and more comprehensive OOD evaluations. (merged into Table 1 and revised the experimental setting section)
- In response to `GZ1U`, we included evaluations on powerful closed-source models. (updated in Table 1)
- For all reviewers, we added finer-grained analyses, case studies, dataset examples, and clarifications. (New related work added; a performance breakdown included in Section 5.2; dataset examples and additional cases are included in Appendix A.3 and Appendix D.1; future work discussion added to Appendix E)

---

We would like to reaffirm the contributions of this work:

- **ReasonMap-Plus** introduces structured dense annotations and difficulty partitioning to support RL cold-start training with dense reward signals for visual reasoning.
- **RewardMap** proposes a multi-stage RL scheme that progressively builds perception -> reasoning, effectively mitigating sparse reward collapse.
- Extensive experiments demonstrate **consistent improvements** across ReasonMap, ReasonMap-Plus, and six OOD benchmarks, supported by ablations, process-level analyses, and cross-model evaluation.

---

We sincerely thank all reviewers for their constructive feedback and efforts in improving our work. We are also deeply grateful to the Area Chairs for their time and careful evaluation.

*Warm regards,*

The Authors of Submission 397

---

### Meta-Review · Area_Chair_AN8f · 2026-01-08

**Summary:**

This submission proposes RewardMap, a multi-stage RL recipe to mitigate sparse-reward collapse for fine-grained visual reasoning in MLLMs, and introduces ReasonMap-Plus, a dense, difficulty-partitioned auxiliary dataset on transit maps. Across reviewers, the main concerns that tempered enthusiasm were the incremental conceptual novelty (largely combining known ingredients like curriculum learning, reward shaping, and difficulty weighting), the practical cost–benefit question, and generalization/scope (initially centered on Qwen2.5-VL and a map-heavy domain). Additional technical questions targeted reward design clarity and sensitivity, potential dataset leakage controls, and a need for more mechanism/qualitative analysis beyond headline metrics.

**Reviewer Concerns:**

The rebuttal and discussion substantially improved clarity and empirical support. The authors directly addressed generalization concerns by adding results on another model family and by reporting stronger reference baselines from closed-source models (e.g., GPT-4o / GPT-5) on ReasonMap/ReasonMap-Plus, which helps contextualize benchmark difficulty and ceilings. They also strengthened the technical narrative by clarifying how sample-level difficulty weights interact with group-based advantage computation, detailing what “multi-stage” means beyond simple easy-to-hard sorting, and providing concrete process-level annotation examples for dataset construction. Importantly, they added ablations over key reward hyperparameters and expanded baselines, as well as more case studies/training-dynamics discussion and an explicit statement about map-level train/test splits to mitigate leakage concerns. That said, some concerns remain only partially resolved: the novelty is still best framed as a careful systems/recipe contribution rather than a new RL principle; the generality of the task-specific detail rewards beyond structured domains like maps/charts is still not fully demonstrated; and the magnitude/consistency of OOD gains remains mixed across tasks, leaving a lingering question about when the added complexity is most justified.

**Reviewer Scores:**

Given the discussion trajectory, I expect 3FQv would likely remain around a 6, as the added ablations, cross-model results, additional baselines, and clearer leakage statement address several technical doubts, while their original novelty reservations would likely persist. GZ1U would likely stay at an 8, since their main asks were addressed and the overall contribution aligns with their positive assessment. owjV has already indicated that the rebuttal addressed most concerns and raised to 6, and I would not expect further movement without substantially broader demonstrations beyond structured domains. ekjh already updated to 4 after added experiments and analysis; I expect they would likely remain at 4 because their core critique about incremental novelty and reward-shaping generality is only partially mitigated, even though the new ablations and task-wise breakdown make the empirical story more credible.

---

### Decision · Program_Chairs · 2026-01-26

Accept (Poster)